# Sequence-dependent material properties of biomolecular condensates and their relation to dilute phase conformations

Dinesh Sundaravadivelu Devarajan [1] ✉, Jiahui Wang[1], Beata Szała-Mendyk [1], Shiv Rekhi[1], Arash Nikoubashman [2,3,4], Young C. Kim[5] & Jeetain Mittal [1,6,7] ✉

Material properties of phase-separated biomolecular condensates, enriched with disordered proteins, dictate many cellular functions. Contrary to the progress made in understanding the sequence-dependent phase separation of proteins, little is known about the sequence determinants of condensate material properties. Using the hydropathy scale and Martini models, we computationally decipher these relationships for charge-rich disordered protein condensates. Our computations yield dynamical, rheological, and interfacial properties of condensates that are quantitatively comparable with experimentally characterized condensates. Interestingly, we find that the material properties of model and natural proteins respond similarly to charge segregation, despite different sequence compositions. Molecular interactions within the condensates closely resemble those within the single-chain ensembles. Consequently, the material properties strongly correlate with molecular contact dynamics and single-chain structural properties. We demonstrate the potential to harness the sequence characteristics of disordered proteins for predicting and engineering the material properties of functional condensates, with insights from the dilute phase properties.

Material properties of biomolecular condensates play a key role in the proper execution of different biological functions, e.g., cell division[1], selective autophagy[2,3], gene regulation[4,5], and nuclear-cytoplasmic shuttling[6]. In general, the material state of condensates ranges from reversible liquid-like assemblies with functional relevance to irreversible solid-like assemblies with pathological consequences[7–10]. The viscous nature (liquidity) of the condensates formed via phase separation enables them to act as dynamic assemblies, which exchange molecules with the surrounding environment, and dissolve as required, in response to physiological cues[11,12]. To better understand the spatiotemporal evolution of condensates, researchers have recently started characterizing their mesoscopic material properties

such as diffusion coefficient, viscosity, viscoelasticity, and surface tension through experiments[13–17]. However, much remains unknown, especially how the protein sequence dictates these different material properties.

Intrinsically disordered proteins (IDPs) or regions (IDRs) are deemed essential for condensate formation[18–21], which across proteomes, typically consist of a high fraction of charged residues[22,23]. Alterations within an IDP sequence, e.g., due to mutation or post-translational modification, can lead to changes in the intra- and inter-molecular interactions, which can impact both the phase behavior and material properties of the condensates formed or can provide the ability to modulate them independently[24]. For example, changes in the

[1]Artie McFerrin Department of Chemical Engineering, Texas A&M University, College Station, TX 77843, USA. [2]Leibniz-Institut für Polymerforschung Dresden e.V., Hohe Straße 6, 01069 Dresden, Germany. [3]Institut für Theoretische Physik, Technische Universität Dresden, 01069 Dresden, Germany. [4]Cluster of Excellence Physics of Life, Technische Universität Dresden, 01062 Dresden, Germany. [5]Center for Materials Physics and Technology, Naval Research Laboratory, Washington, DC 20375, USA. [6]Department of Chemistry, Texas A&M University, College Station, TX 77843, USA. [7]Interdisciplinary Graduate Program in Genetics and Genomics, Texas A&M University, College Station, TX 77843, USA. ✉e-mail: dineshsd@tamu.edu; jeetain@tamu.edu

electrostatic interactions of the highly disordered full-length FUS protein, either by a specific point mutation to glutamic acid[25] or by phosphorylating serine residues[26] within the prion-like domain, did not significantly influence its phase separation behavior. However, such alterations had a contrasting effect on the material state of its condensates: the mutated variant led to a dynamically arrested condensate[25], while the phosphorylated variant prevented condensate aggregation[26]. In addition, poly-arginine condensates exhibited significantly reduced dynamics (100-fold higher viscosity) than poly-lysine condensates[27], but replacing arginine with lysine residues in artificial IDPs resulted only in a two-fold decrease in viscosity[24]. Despite the significant progress made regarding the sequence determinants of phase separation[24,28–34], identifying how the IDP sequence modulates its condensate material properties remains an important open question[35]. Further, establishing the molecular basis of sequence-encoded material properties will provide insights for condensate biology and for designing synthetic condensates with tunable biophysical characteristics.

Deciphering the relationship between sequence features and material properties of charged IDPs is the major goal of this study. To achieve this, we make use of the patterning of oppositely charged residues within polyampholytic sequences, which has been shown to dictate the conformations and phase behavior of model IDPs and naturally occurring IDRs[22,32,36,37]. Recently, alternating charge blockiness within certain IDRs was found to be crucial for their selective partitioning into condensates for transcriptional regulation[38], suggesting a role for the condensate interfaces[34,39]. We perform coarse-grained molecular dynamics (MD) simulations using the hydropathy scale (HPS) and Martini models to investigate how changes to the charge patterning within IDP sequences influence their condensate material properties such as diffusion coefficient, viscosity, and surface tension. We use model proteins consisting of negatively charged glutamic acid (E) and positively charged lysine (K) residues as well as charge-rich naturally occurring proteins such as the LAF1's RGG domain (hereafter referred to as LAF1) and the DDX4's N-terminal domain (hereafter referred to as DDX4). Our results reveal that the effect of charge segregation on different material properties of the condensates is highly similar between the model proteins and natural proteins. In general, we find that charge blockiness leads to a slow-down in the condensate dynamics, which originates from pronounced electrostatic attraction between the oppositely charged residues in the condensed phase. Further, we show that the molecular interactions within the condensates closely resemble their single-chain interactions. Consequently, we find the material properties of IDP condensates to be strongly correlated with their dense phase dynamics of contact formation and breakage between the oppositely charged residues and with their single-chain structural properties. Our findings demonstrate a way to utilize the sequence-level features of charge-rich IDPs for modulating their condensate material properties and to infer such properties based on the single-chain conformations, which can be characterized via fully atomistic simulations or experiments.

## Results

The effect of charge patterning on the thermodynamic phase behavior of disordered proteins with varying sequence characteristics such as the chain length ($N$), the fraction of charged residues (FCR), and the net charge per residue (NCPR) is a well-studied problem[32,40–42]. For example, Lin and Chan[40] demonstrated that charge patterning in polyampholytic disordered proteins with fixed $N$ altered their critical temperature $T_c$ for phase separation such that its changes were synonymous with that of their single-chain size. Later, Dignon et al.[41] showed that $T_c$ and the single-chain coil-to-globule transition temperature $T_\theta$ are strongly correlated for a diverse set of natural disordered proteins, which included sequences with different charge patterning. However, the effect of charge patterning on the

condensate material properties (e.g., viscosity and surface tension) of proteins with different sequence lengths and charge content is not well understood. To investigate this aspect, we employed the model E-K variants[37] with $N = 50$ residues, and the naturally occurring LAF1 and DDX4 sequence variants with $N = 168$ residues and $N = 236$ residues, respectively (see Supplementary Table 1 for the amino acid sequences and "Methods" section for model details). All model sequences had FCR=1 and NCPR=0 as they consisted of an equal fraction of oppositely charged E and K residues. In the case of the investigated natural protein variants, all LAF1 sequences had FCR=0.262 and NCPR=0.024, whereas all DDX4 sequences had FCR=0.288 and NCPR=−0.017.

The degree of charge segregation in the sequences was quantified using the sequence charge decoration (SCD) parameter[43,44]. Smaller values of SCD correspond to a more charge-segregated sequence. To compare different polyampholyte compositions and lengths, we defined a normalized SCD (nSCD) parameter that is scaled by the maximum and minimum SCD values achievable for each of the E-K, LAF1, and DDX4 sequence compositions. The values of this parameter lie in the interval $0 \leq nSCD \leq 1$, with the lower and upper bounds corresponding to a perfectly alternating sequence and diblock sequence, respectively[37,45]. The sequences and their nSCD value are shown in Fig. 1a. Note that the LAF1 sequence with nSCD=0.010 and the DDX4 sequence with nSCD=0.021 are the wild-type (WT) sequences. We characterized the diffusion coefficient, viscosity, and surface tension of the condensates by simulating the protein chains in bulk systems and phase coexistence systems (Fig. 1b), respectively, using the HPS model (see "Methods" section for model and simulation details). All results for the E-K and DDX4 sequence variants were obtained at a fixed temperature of $T = 300$ K. The results of LAF1 sequence variants were obtained at $T = 280$ K as the critical temperature of its WT obtained from our coarse-grained (CG) model[46] (Supplementary Fig. 1) is $T \approx 290$K.

To investigate the ability of our sequence variants to phase separate, we simulated them in a cubic simulation box at a constant pressure of $P = 0$ atm, which allowed the sequences to reach their preferred dense phase concentration $\rho$. We found that the E-K variant with nSCD=0 did not form a stable dense phase, indicating that the intermolecular attraction between oppositely charged residues was too weak due to the perfectly alternating arrangement of charges in their sequence[40]. In addition, the most well-mixed LAF1 (nSCD=0.003) and DDX4 (nSCD=0.002) variants also did not form a stable dense phase. However, other E-K variants, and all the LAF1 and DDX4 sequences with nSCD≥0.01 formed a dense phase. In fact, we found that the condensates were stable over a wide range of temperatures, as indicated by the increase in critical temperature with increasing charge segregation for the model proteins[40,41] and natural proteins (Supplementary Fig. 1; see "Methods" section). For the sequences that phase separated, we found that $\rho$ monotonically increased with increasing nSCD (i.e., increasing charge segregation) in a nonlinear fashion: the systems became significantly denser for a small increase in nSCD below 0.2 followed by a slower increase in $\rho$ for larger nSCD (Fig. 1c). Further, we found the concentrations of LAF1 and DDX4 variants to be significantly lower compared to the E-K variants. This meant that our sequences facilitated the examination of charge patterning effects over a wide concentration range, which also covers the concentrations of well-studied disordered protein condensates, e.g., low-complexity domains of FUS and hnRNPA2[20,21]. Note that the sequence variants with nSCD<0.01 did not form a stable dense phase in the phase coexistence simulations either, while the dense phase concentrations of the other sequences obtained from bulk and phase coexistence simulations were nearly identical (Supplementary Fig. 2).

Given that the role of counterions and explicit water particles could play an important role in dictating the dynamics and rheology of charge-rich IDP condensates, we also simulated all the E-K sequences using the Martini force field (Supplementary Fig. 3a; see "Methods"

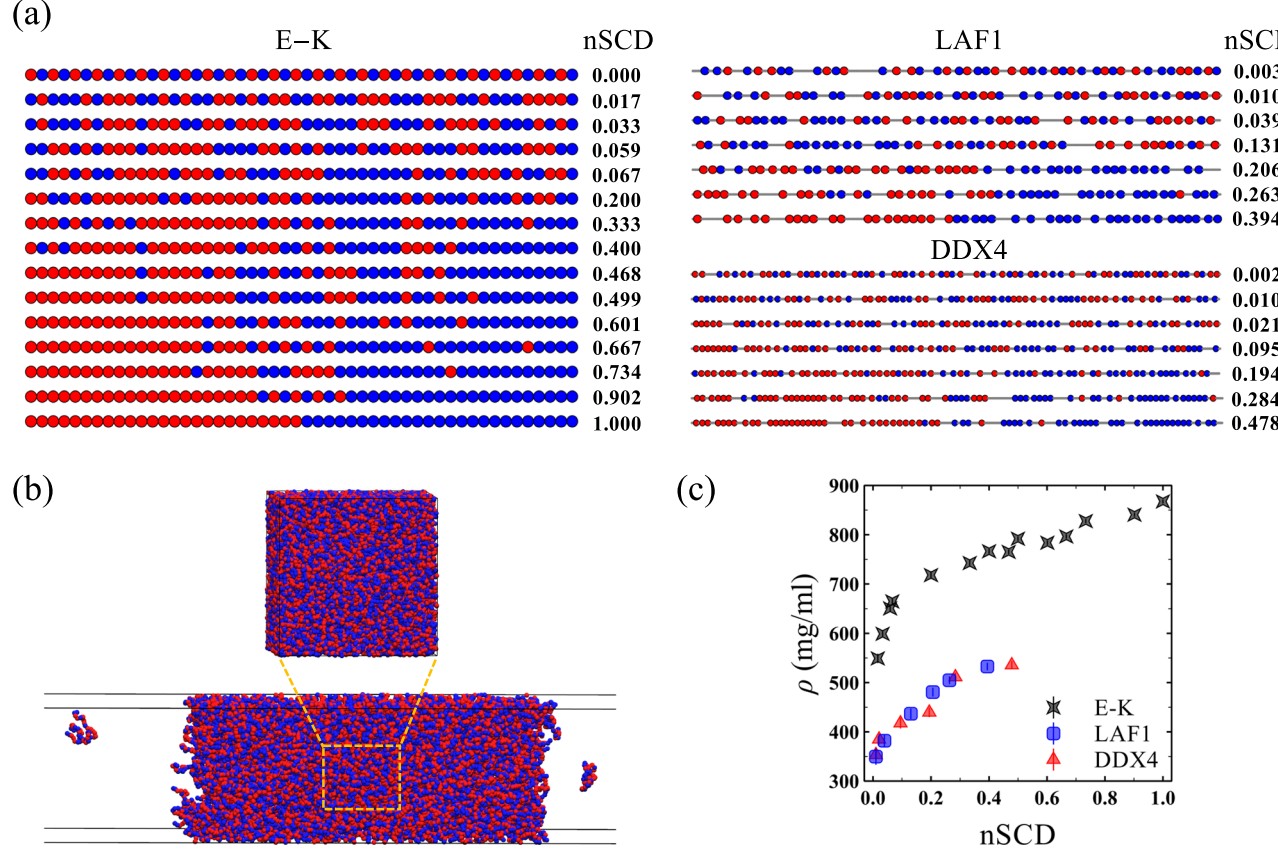

**Fig. 1 | Model and natural IDP sequences, and setup for computing their condensate material properties. a** E-K (chain length, $N = 50$), LAF1 ($N = 168$), and DDX4 ($N = 236$) sequences with their normalized SCD (nSCD) parameter. The LAF1 sequence with nSCD=0.010 and the DDX4 sequence with nSCD=0.021 are the wild-type sequences. The negatively and positively charged residues are shown as red and blue beads, respectively. The uncharged residue positions in LAF1 and DDX4 are shown as gray line segments connecting the charged residues. **b** Simulation snapshots of one of the E-K sequences in a cubic simulation box for characterizing its dense phase dynamical and rheological properties, and in a slab geometry for characterizing its interfacial property between the dense and dilute phases. **c** Dense phase concentration $\rho$ as a function of nSCD for all sequences. Error bars in (**c**) are standard deviations about the mean. The mean values are obtained as an average over $n = 2$ independent constant pressure $P = 0$ atm simulations. Source data are provided as a Source Data file.

section for model and simulation details). Similar to the HPS model, we found that the perfectly alternating E-K variant with nSCD=0 did not phase separate in the Martini model. For the other E-K sequences, we found that the dense phase concentration $\rho$ for the protein increased with increasing nSCD (i.e., increasing charge segregation), similar to that seen in the HPS model (Supplementary Fig. 3b). The absence of explicit solvent in the HPS model resulted in a more concentrated dense phase compared to the Martini model. While the concentration of water decreased monotonically with increasing nSCD in the Martini model, it remained higher than the protein concentration for all E-K sequences; this behavior is similar to that observed from all-atom simulations of condensates formed by natural disordered proteins such as the FUS low-complexity domain and LAF1 RGG domain[21]. These observations are consistent with the expectation that increasing charge blockiness within the sequence should lead to pronounced electrostatic attraction between the oppositely charged residues, resulting in a denser protein phase.

## Charge segregation in IDP sequences leads to monotonic changes in their dense phase material properties

Since our model and natural protein sequences have different sequence compositions, we asked whether the segregation of charges would have a similar effect on the dynamical, rheological, and interfacial properties of their phase-separated condensates. We investigated this aspect by first characterizing the translational motion of the protein chains in the dense phase. In experiments, the chain motion is

often estimated from fluorescence recovery after photobleaching (FRAP)[47]. In simulations, we can directly measure the chain motion by monitoring the mean square displacement of their residues as a function of time

$$\mathrm{MSD}(t) = \left\langle \left[ \boldsymbol{r}_i(t) - \boldsymbol{r}_i(0) \right]^2 \right\rangle, \tag{1}$$

where $\boldsymbol{r}_i(t)$ is the position of residue $i$ at time $t$. To avoid chain end effects, we excluded 20 residues on either end of a chain for computing the MSD. We found that the residues of all sequences exhibited the same ballistic motion (MSD $\propto t^2$) at short times (Supplementary Fig. 4), which was followed by a sub-diffusive motion[48] (MSD $\propto t^{1/2}$) of residues at intermediate times, before they eventually showed normal diffusion (MSD $\propto t$) at sufficiently long times. The proportional scaling MSD $\propto t$ indicated that the condensates of all sequences exhibited largely viscous behavior under the conditions of this study. Most importantly, the MSD showed a strong variation with charge segregation for all sequences in the diffusive regime. To quantify this trend, we computed the diffusion coefficient $D$ by applying the relationship MSD $= 6Dt$ at long times for all sequences. This way of computing $D$ from the MSD of the residues in a chain accurately reflects its translational motion in the dense interior of the condensates (see "Methods" section)[49].

To account for the chain length dependence of the diffusion coefficient, we report $D^*$ for the E-K variants obtained after normalizing $D$ by that computed for the reference E-K sequence with

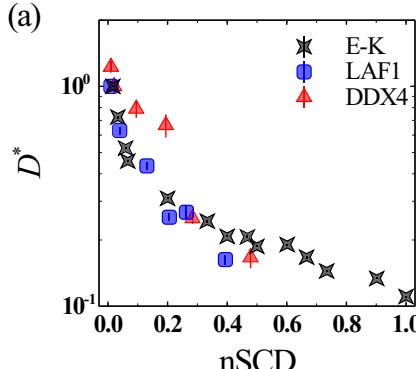

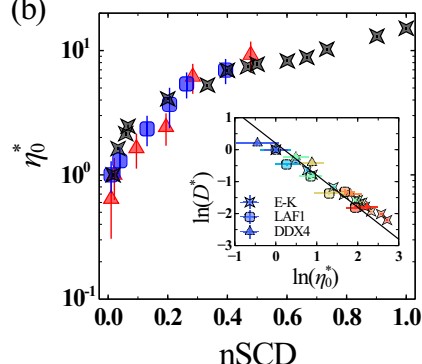

**Fig. 2 | Condensate dynamics and rheology show similar response to sequence charge segregation, despite distinct IDP compositions.** Normalized material properties of the dense phase as a function of nSCD for the E-K, LAF1, and DDX4 sequences: **a** diffusion coefficient $D^*$ and **b** zero-shear viscosity $\eta_0^*$. For the E-K sequences, $D^*$ and $\eta_0^*$ are obtained after normalizing $D$ and $\eta_0$ by those computed for the reference E-K sequence with nSCD=0.017. For the LAF1 and DDX4 sequences, $D^*$ and $\eta_0^*$ are obtained after normalizing $D$ and $\eta_0$ by those computed for their respective wild-type sequences. The inset shows the inversely proportional correlation between $D^*$ and $\eta_0^*$ for all sequences, with the solid line corresponding to the Stokes–Einstein type relation $D^* = 1/\eta_0^*$. The symbol color, ranging from blue to red, indicates increasing nSCD. Error bars in all panels indicate standard errors about the mean. The mean values are obtained as an average over $n = 2$ independent simulations, with each replica divided into 3 independent blocks to estimate error bars. Source data are provided as a Source Data file.

nSCD=0.017, which was the lowest nSCD sequence that phase separated under the conditions of this study (Fig. 2a). Similarly, the $D^*$ values for the LAF1 and DDX4 variants were attained after normalizing $D$ by those computed for their respective WT sequences, which exhibited a well-mixed charge distribution with a similar nSCD value as the reference E-K sequence. We found that $D^*$ monotonically decreased with increasing nSCD for both the model and natural proteins (Fig. 2a). Interestingly, the rate of decrease in $D^*$ was also similar, despite their very different sequence compositions. The reduced diffusion coefficient of charge-segregated sequences is consistent with a previous simulation study of similar polyampholyte sequences[50], and has also been validated experimentally for the LAF1 WT sequence and its charge-shuffled variant[32]. From these observations, we concluded that the segregation of charges leads to a slowdown in the dynamics of condensates formed by charge-rich IDPs.

The material state of the condensates can also be characterized through rheological properties such as the shear viscosity. For this purpose, we used the non-equilibrium MD (NEMD) simulation technique[51–53] (see "Methods" section for simulation details) for subjecting the dense phase of the protein chains to steady shear at different shear rates $\dot{\gamma}$, and measured the resulting shear stress component $\tau_{xy}$. We then determined their shear viscosity

$$\eta = -\frac{\langle \tau_{xy} \rangle}{\dot{\gamma}}, \quad (2)$$

and plotted it as a function of $\dot{\gamma}$ (Supplementary Fig. 5). We found near-constant $\eta$ values at sufficiently low shear rates for all sequences, indicative of their Newtonian liquid-like behavior, similar to that seen for experimentally characterized disordered protein condensates[18,24,32]. Similar to the MSD, $\eta$ depended on the degree of charge segregation within the sequence. To illustrate this dependence, we considered the zero-shear viscosity $\eta_0$ of all sequences, which is obtained as an average of the $\eta$ values from the Newtonian plateau observed at low shear rates (Supplementary Fig. 5). Again, we normalized $\eta_0$ by that obtained for the reference sequences (i.e., E-K sequence with nSCD=0.017 for the E-K variants, and respective WT sequences for the LAF1 and DDX4 variants), which we denote as $\eta_0^*$ (Fig. 2b). We found that $\eta_0^*$ monotonically increased with increasing nSCD. Further, similar to $D^*$, $\eta_0^*$ of all proteins with different sequence characteristics collapsed onto a master curve for the entire range of nSCD. This observation indicated that the interchain electrostatic interactions get stronger as a result of charge blockiness within the sequence of charge-rich IDPs.

The fact that $D^*$ and $\eta_0^*$ depended on nSCD and that the changes in these quantities were highly similar between the E-K, LAF1, and DDX4 sequence variants, prompts the question: can a simple Stokes–Einstein type relation (i.e., $D^* = 1/\eta_0^*$) capture the changes in these quantities of the charge-patterned sequences? Recent simulation studies have shown that the Stokes–Einstein type relation holds for model associative proteins with varying interaction strength[54,55]. Indeed, we found that the changes in $D^*$ and $\eta_0^*$ in response to charge patterning closely followed the relation $D^* = 1/\eta_0^*$ (inset of Fig. 2b). This behavior was also true in the case of $D^*$ and $\eta_0^*$ values obtained using the Martini model (Supplementary Fig. 3c; see "Methods" section), indicating that the observed behavior is model independent. Thus, our results demonstrate the feasibility of predicting these quantities from one another as the charge pattern within the IDPs gets altered. Such a correlation between $D^*$ and $\eta_0^*$ also highlights that the interchain electrostatic interactions simultaneously dictate both the chain dynamics and rheology of the dense phase of charge-rich IDPs.

**Interfacial and dense phase properties of charge-rich IDP condensates obtained from simulations and experiments are comparable**

Our analysis of the material properties in the dense phase revealed that the IDPs exhibited liquid-like characteristics, though these properties were found to change in a sequence-dependent manner due to changes in the intermolecular interactions within the condensate. Another determinant of condensate liquidity is the surface tension $\gamma$ at the interface between the dense and dilute phases, which together with the zero-shear viscosity $\eta_0$ of the dense phase can dictate the speed of droplet fusion[15,16,56]. We determined $\gamma$ using the Kirkwood–Buff relation[57]

$$\gamma = \frac{L_z}{2} \left\langle P_{zz} - \frac{P_{xx} + P_{yy}}{2} \right\rangle, \quad (3)$$

where $L_z$ is the edge length of the simulation box in the $z$-direction, $P_{ii}$ is the pressure tensor component in a given direction, and the factor of $1/2$ accounts for the two interfaces present in the slab geometry of the phase coexistence simulations (see Fig. 1b). We use a normalized quantity $\gamma^*$ (Fig. 3a), obtained after normalizing $\gamma$ by the corresponding value of the same reference sequences used in getting $D^*$ and $\eta_0^*$ of the

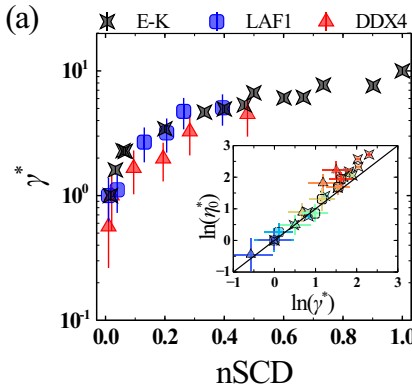
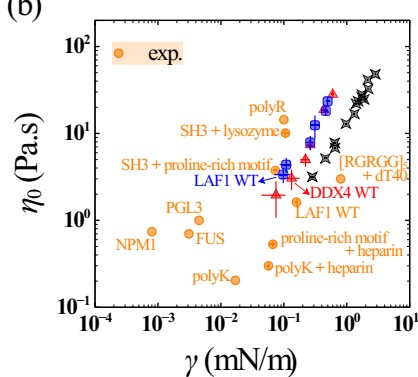

**Fig. 3 | Sequence-dependent interfacial and dense phase properties of IDP condensates exhibit analogous changes. a** Normalized surface tension $\gamma^*$ as a function of nSCD for the E-K, LAF1, and DDX4 sequences. The value of $\gamma^*$ is obtained after normalizing $\gamma$ by those computed for the same reference sequences used in the normalization of $D$ and $\eta_0^*$ in Fig. 2. The inset shows the correlation between $\eta_0^*$ and $\gamma^*$ for all sequences, with the solid line corresponding to $\eta_0^* = \gamma^*$. The symbol color, ranging from blue to red, indicates increasing nSCD. **b** Comparison of zero-shear viscosity $\eta_0$ and surface tension $\gamma$ obtained for the simulated E-K, LAF1, and DDX4 dense phases with those measured for a wide range of protein condensates through experiments. Error bars in all panels indicate standard errors about the mean. The mean values are obtained as an average over $n = 2$ independent simulations, with each replica divided into 3 independent blocks to estimate error bars. Source data are provided as a Source Data file.

E-K, LAF1, and DDX4 sequences. We found that the segregation of charges increased $\gamma^*$, much like its effect on $\eta_0^*$. The values of $\gamma^*$ and $\eta_0^*$ for all IDPs were positively correlated (inset of Fig. 3a), highlighting that both these quantities can be simultaneously modulated by altering the charge patterning of charge-rich IDPs. Taken together, our results revealed that large changes in the material properties of charge-rich IDP condensates can already occur solely through the sequence charge patterning, at constant external conditions such as temperature and salt concentration.

Having assessed the interfacial and dense phase properties of charge-rich IDP condensates using our physics-based CG model, we next investigated if they lie in the expected range of experimentally characterized material properties of other naturally occurring protein condensates (Fig. 3b). It is known that CG models can significantly underpredict viscosity as compared to experimental measurements[58], which is primarily due to the smoothening of the free energy landscape[59] and inaccurate solvent frictional effects. Thus, to account for these differences and enable a meaningful comparison between simulations and experiments, it is crucial to establish the connection between their relevant timescales. For this purpose, we derived timescales based on a mesoscopic quantity, specifically the translation diffusive motion of a single protein chain in dilute solution[60–62], for interpreting the dynamical and rheological properties of the condensed phase obtained from our implicit-solvent CG simulations (see "Methods" section for details on our timescale mapping strategy). We found that the diffusion coefficients $D$ within LAF1 and DDX4 WT dense phases, based on the timescales mapped at the single-chain level in solution, were in the comparable range to those obtained from FRAP experiments on the corresponding in vitro droplets[32,36] (Supplementary Table 2). Remarkably, the values of zero-shear viscosity $\eta_0$ and surface tension $\gamma$ of the LAF1 WT sequence obtained from simulations and experiments were quantitatively comparable[15] (see "Methods" section for discussion on the experimental methods used to characterize $\eta_0$ and $\gamma$). In fact, the range of $\eta_0$ and $\gamma$ for the simulated charge-rich IDP condensates were in close proximity to the $\eta_0$ and $\gamma$ values of other protein condensates that are primarily driven by electrostatic interactions[16,27,63,64], namely PGL3, polyR, polyK, polyK +heparin, SH3-targeting proline-rich motif+heparin, and [RGRGG]₅ + dT40 (Fig. 3b). These results demonstrate the ability of our CG model to provide an accurate description of the condensate material properties for a wide range of proteins.

## Material properties of IDP condensates have a strong connection to the microscopic contact dynamics

Our analysis of the material properties of IDP condensates indicated their dependence on the changing nature of molecular interactions as a result of sequence charge patterning. To characterize these changes, we first computed the radial distribution function (RDF) between the oppositely charged residues of each sequence (Supplementary Fig. 6), which we then converted to the potential of mean force (PMF) required for separating them within the dense phase for a given sequence as PMF = $- k_B T \ln(\text{RDF})$[65], where $k_B$ is the Boltzmann constant (Supplementary Fig. 7). We defined the difference in PMF value at the first minimum (corresponding to the bound state) and PMF=0 (corresponding to the unbound state) as the free energy change $\Delta F$ (Fig. 4a). We found $\Delta F > 0$ for the low-nSCD sequences (i.e., nSCD$\lesssim$0.20), indicating the absence of an attractive well for these sequences. However, the segregation of charges decreased $\Delta F$ for the E-K, LAF1, and DDX4 sequences, highlighting the pronounced electrostatic attraction between the oppositely charged residues of charge-rich IDPs. These results provide a molecular mechanistic description of our finding that the charge-segregated sequences exhibit slower dynamics compared to the uniformly charge-patterned sequences (Fig. 2). Further, the implications of sequence charge patterning on the interfacial and dense phase properties (Fig. 3) are in line with previous experiments, which showed that the screening of electrostatic interactions at high salt concentrations resulted in the decrease of the surface tension and viscosity of PGL3 protein droplets[63].

Given that the interaction strength between the oppositely charged residues in the dense phase is highly sensitive to the charge patterning within the IDPs, we next investigated their dynamics of contact formation, which would facilitate a direct comparison with the material properties of IDP condensates. We computed the intermittent contact time autocorrelation[54,55]

$$c(t) = \frac{\langle h(0)h(t)\rangle}{\langle h^2(0)\rangle},\tag{4}$$

with step function $h(t) = 1$ if a pair of oppositely charged residues $i$ and $j$ between two chains were in contact (i.e., if the distance between the pair was less than the cut-off radius of 1.5$\sigma$, where $\sigma$ is the average diameter of residues $i$ and $j$) at time $t = 0$ and at time $t$, irrespective of whether they stayed in contact at intermittent times, and $h(t) = 0$ if not

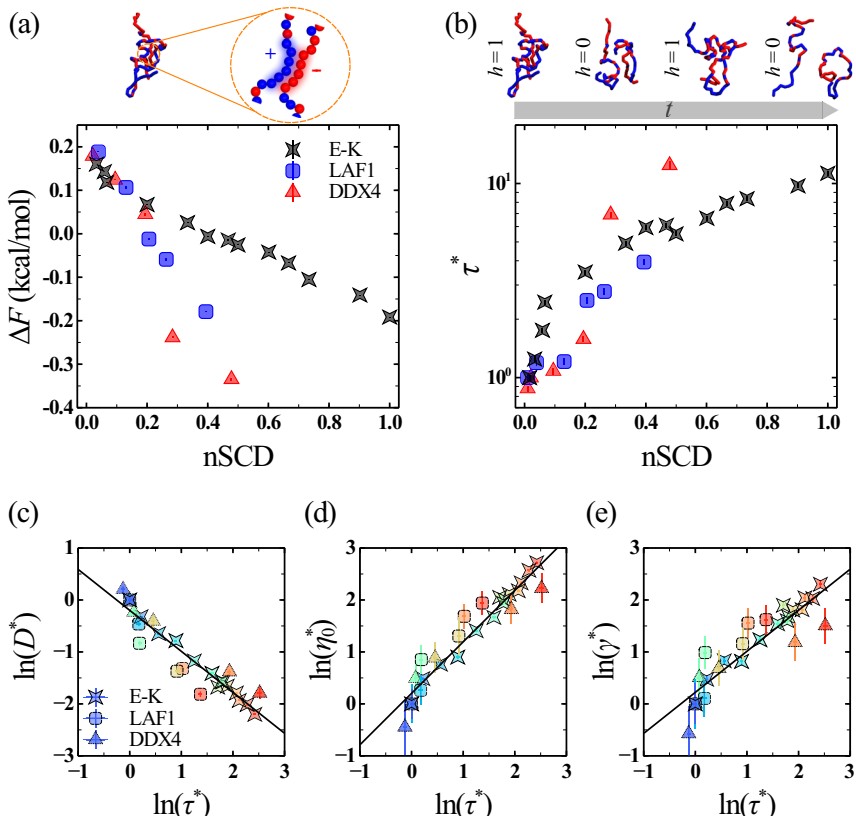

**Fig. 4 | Microscopic contact dynamics is reflective of the condensate material properties. a** Free energy change $\Delta F$ between the bound and unbound states and **b** normalized intermittent contact lifetime $\tau^*$ for the interchain oppositely charged residues as a function of nSCD for the E-K, LAF1, and DDX4 sequences. The values of $\tau^*$ are obtained after normalizing $\tau$ by those computed for the same reference sequences used in the normalization of the material properties as in Fig. 2. In (**b**), an example for the time evolution of the intermittent contact formation ($h(t) = 1$) and breakage ($h(t) = 0$) between a pair of chains for the E-K variant with nSCD=0.468 is also shown. Correlation **c** between $D^*$ and $\tau^*$, **d** between $\eta_0^*$ and $\tau^*$, and **e** between $\gamma^*$

and $\tau^*$ for all sequences. The symbol color, ranging from blue to red, indicates increasing nSCD. The solid lines in (**c**–**e**) correspond to linear fits $\ln(D^*) = -0.79\ln(\tau^*) - 0.20$ with correlation coefficient $R^2 = 0.90$, $\ln(\eta_0^*) = 0.99\ln(\tau^*) + 0.20$ with $R^2 = 0.93$, and $\ln(\gamma^*) = 0.79\ln(\tau^*) + 0.22$ with $R^2 = 0.83$, respectively. Error bars in all panels indicate standard errors about the mean. The mean values are obtained as an average over $n = 2$ independent simulations, with each replica divided into 3 independent blocks to estimate error bars. Source data are provided as a Source Data file.

(Fig. 4b). This definition of autocorrelation accounts for the contacts between residues that get broken and reformed again, thus measuring the duration for which residues from two chains remain in the same vicinity. We found that $c$ (averaged over all interchain pairs of oppositely charged residues) decayed differently with increasing charge segregation for the model and natural proteins (Supplementary Fig. 8). To quantify the variations in $c$, we computed the intermittent contact lifetime $\tau$ (Fig. 4b) by integrating $c$ up to the time for which it reached a value of 0.05[54,55,66]. Similar to the representation of different material properties, we normalized $\tau$ by the corresponding value of the reference sequences, giving rise to $\tau^*$.

Interestingly, $\tau^*$ increased with increasing nSCD, in line with the observed decrease in $\Delta F$, for the E-K, LAF1, and DDX4 sequences, further highlighting the role of protein sequence via charge distribution. Given that the different material properties of IDPs also depended on the charge distribution along the sequence, we hypothesized that the dynamics of contact formation and breakage between the oppositely charged residues could predominantly contribute to such condensed phase properties. To validate this hypothesis, we tested for a correlation between $\tau^*$ and each of the different material properties $D^*$, $\eta_0^*$, and $\gamma^*$ by plotting a logarithmic version of these quantities against each other (Fig. 4c–e). In this representation, the simulation data of each material property fell on a master curve that followed a power-law relation with the contact dynamics, as quantified by the correlation coefficients ranging from $R^2 = 0.83$ to $R^2 = 0.93$. These

observations validated our hypothesis that a longer contact lifetime between the oppositely charged residues contributed to a lower diffusion coefficient, a higher viscosity, and a higher surface tension of the charge-rich IDP condensates. Further, the dynamics of contact formation and breakage were highly transient (Supplementary Fig. 8), consistent with the recent finding that the local interactions were exceedingly rapid in the phase-separated condensates formed by two highly charged polyelectrolytic IDPs like histone H1 and prothymosin $\alpha$[66]. This rapid dynamics of formation and breakage of contacts between oppositely charged residues, in conjunction with its correlation to the material properties, implies that short equilibrium simulations for characterizing the contact dynamics can help to infer the condensed phase properties of charge-rich IDPs.

## Condensed phase material properties and single-chain structural properties of charged disordered proteins are strongly correlated

It is well-established that the segregation of charges within the polyampholyte sequences alters their single-chain interactions[22,37,67]. Having shown that the interactions in the dense phase are highly sensitive to the charge patterning within the polyampholytic IDPs, we next investigated whether they shared similar features with the corresponding self-interactions at the single-chain level. For this purpose, we computed the probability of a pair of residues $i$ and $j$ to be in contact as $P = \langle n_{ij} \rangle$, with $n_{ij} = 1$ if the distance between the pair was less

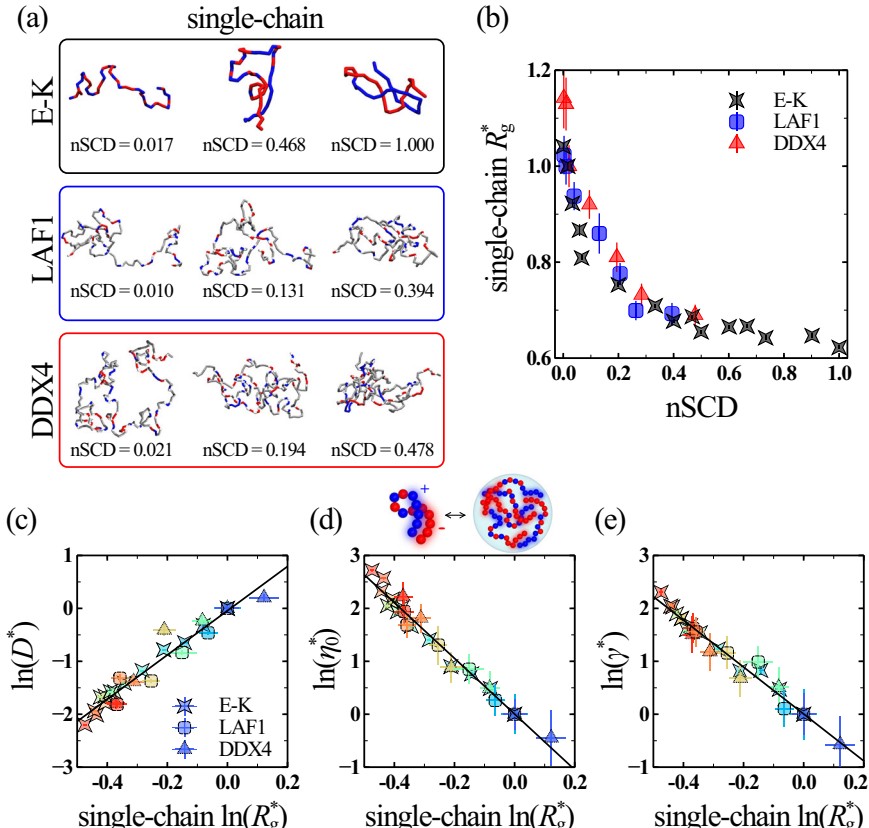

**Fig. 5 | Dilute phase conformations of IDPs inform their condensate material properties. a** Representative single-chain conformations chosen close to the mean radius of gyration $R_g$ of a given nSCD for select E-K, LAF1, and DDX4 sequences. Negatively charged, positively charged, and neutral residues are shown in red, blue, and gray colors, respectively. **b** Normalized single-chain radius of gyration $R_g^*$, obtained after normalizing $R_g$ by those computed for the same reference sequences used in the normalization of the material properties as in Fig. 2, as a function of nSCD for all sequences. Correlation **c** between $D^*$ and $R_g^*$, **d** between $\eta_0^*$ and $R_g^*$, and **e** between $\gamma^*$ and $R_g^*$ for all sequences. The symbol color, ranging from blue to red, indicates increasing nSCD. The solid lines in (**c**–**e**) correspond to the linear fits $\ln(D^*) = 4.18\ln(R_g^*) - 0.04$ with correlation coefficient $R^2 = 0.94$, $\ln(\eta_0^*) = -5.26\ln(R_g^*) + 0.005$ with $R^2 = 0.98$, and $\ln(\gamma^*) = -4.43\ln(R_g^*) + 0.005$ with $R^2 = 0.97$, respectively. Error bars in (**b**–**e**) are standard errors about the mean. The mean values are obtained as an average over $n = 2$ independent simulations, with each replica divided into 3 independent blocks to estimate error bars. Source data are provided as a Source Data file.

than the cut-off radius of $1.5\sigma$ and $n_{ij} = 0$ if not. We then summed over the contact probabilities $\sum P$ of a residue $i$ with all other oppositely charged residues or like charged residues or uncharged residues in the dense phase and within a single chain for a given sequence (see "Methods" section). By comparing the $\sum P$ values in the dense phase with those obtained at the single-chain level, we found that they were well-correlated across the entire nSCD range for the E-K, LAF1, and DDX4 sequences (Supplementary Figs. 9–11). These observations indicated that similar molecular interactions are at play for a charge-rich IDP in the dense phase as well as in the ultra-dilute phase[40,66]. This similarity is the basis for using the single-chain conformations (quantified via the radius of gyration) as an indicator of the propensity of model IDP solutions and naturally occurring IDP solutions to phase separate into a condensate (quantified via the critical temperature or the saturation concentration)[40,41]. In particular, our previous work[41] has established that single-chain simulations of coil-to-globule transitions are powerful tools to interpret the phase behavior of protein sequences. Subsequent studies have shown that the molecular driving forces behind single-chain conformations and condensate formation can be decoupled when the NCPR of the sequence keeps deviating from zero[42] or the hydrophobic residues are clustered in the sequence[33,68,69]. However, such deviations were not observed irrespective of the patterning of the oppositely charged residues in IDP sequences[40,41].

Based on the above considerations, we next asked whether the single-chain properties of charge-rich IDPs with different NCPR values and with varying distributions of oppositely charged residues could serve as an indicator of the trends in the material properties of the condensates formed by them. We characterized the single-chain conformations of the E-K[37], LAF1, and DDX4 sequence variants through their radius of gyration $R_g = \langle G_1 + G_2 + G_3 \rangle^{1/2}$, which we computed from the eigenvalues $G_i$ of the gyration tensor (Supplementary Fig. 12 and Fig. 5a)[37]. In line with previous computational studies[22,37,50], we found that $R_g$ decreased with increasing nSCD for the model proteins and natural proteins, because of pronounced electrostatic attraction between the oppositely charged residues within the sequence. Further, $R_g$ of the natural proteins were significantly larger than the model proteins because of their longer chain lengths. Thus, to eliminate the chain length dependence of $R_g$, we normalized it for each sequence variant by the corresponding value of the reference sequences to define $R_g^*$ (Fig. 5b), similar to the normalized representation of the different material properties. Analogous to the trends seen for the different material properties, we found that the rate of decrease in $R_g^*$ in response to the segregation of charges within the sequences of different compositions were nearly identical, thus substantiating the predominant role of electrostatic interactions for their single-chain properties. Next, we tested for the correlation between $R_g^*$ and each of the different material properties $D^*$, $\eta_0^*$, and $\gamma^*$ of charge-rich IDPs (Fig. 5c–e). We found that the simulation data of all sequences

collapsed onto a master curve, resulting in a strong power-law correlation between the single-chain size and the different condensed phase properties, with correlation coefficients $R^2 \approx 0.96$. Interestingly, we found that the magnitude of the power-law exponent ~5 describing the correlation between the different material properties and single-chain $R_g$ was highly similar to that obtained for the correlation between the critical temperature and single-chain $R_g$ of charge-rich IDPs[40]. Further, we found that the coupling between the single-chain conformations and the condensate material properties persists even when they are computed based on two different hydropathy scales (Supplementary Fig. 13) or using the Martini model (Supplementary Fig. 3d; see "Methods" section), indicating that the observed correspondence does not depend on the model choice. In general, the higher the single-chain compaction of IDPs induced by sequence charge segregation, the slower the dynamics of the condensates formed by them. These findings highlight that the extent of single-chain compactness can serve as a useful indicator of the material state of IDPs, emergent upon their phase separation to form condensates.

## Discussion

Several studies exist on the sequence-dependent phase separation of IDPs, but the material state of IDP condensates and its dependence on sequence is only beginning to be explored. Investigating the sequence determinants of the material properties of disordered protein condensates is exciting for two reasons: (1) it can help predict how sequence alterations perturb the intermolecular interactions, thereby affecting the condensate material properties that could modify biological function, and (2) it will enable the development of sequence design principles for engineering condensates with targeted material properties. Using a physics-based transferable CG framework (HPS model)[46,70] that has been successfully applied to decipher the sequence-phase behavior relationship of IDPs, we performed a molecular-level investigation of how the sequence of the charged disordered proteins dictates their condensate material properties. The fact that the specific arrangement of the charges within an IDP sequence can influence its conformations[22,37] and phase behavior[32,40] makes it an attractive feature to decipher the sequence-material property relationship of IDPs. Thus, we employed the charge-patterned variants of a model protein consisting of an equal number of glutamic acid and lysine residues as well as the charge-patterned variants of disordered regions of two naturally occurring proteins, namely LAF1 and DDX4. We characterized the condensates formed by these protein variants through the equilibrium and non-equilibrium MD simulations using the HPS and Martini models. Our analysis revealed that charge patterning led to monotonic changes in different material properties despite their very different sequence compositions: with increasing charge segregation, the diffusion coefficient of the protein chains within the dense phase decreased, while the dense phase viscosity and the surface tension at the condensate-water interface increased. Further, the rate of change of these material properties with changing charge distribution was nearly identical between the model and natural proteins, thus highlighting the interdependence between these properties across a wide range of sequence compositions. These observations highlight that sequence charge patterning can help modulate the material properties within the dense phase as well as at the interface of charge-rich IDP condensates, without changing external conditions such as temperature and salt concentration. Further, our results will serve as a baseline to investigate the use of different charge patterning parameters to predict the material properties of protein condensates over a broader sequence space, similar to the literature studies[71] on the usefulness and limitations of such parameters for predicting the phase separation propensity of IDPs.

The sensitivity of the material properties to the segregation of charges in the sequence suggested that the interchain electrostatic interactions played a dominant role in influencing them. This hypothesis was supported by our finding that the intermittent lifetime of contacts between the oppositely charged residues increased when the charges were segregated for both the model and natural proteins. This behavior is consistent with associative polymer theories[72–74], which imply that stronger interchain interactions lead to slower condensate dynamics[75]. However, to the best of our knowledge, it has not been shown whether a universal correlation exists between the *microscopic* contact dynamics and the *mesoscopic* material properties with changes in sequence charge patterning. Interestingly, we found a strong universal correlation between the lifetime of noncovalent bonds formed by the oppositely charged residues and the different material properties of all investigated sequences, despite their different sequence characteristics. However, we found the dynamics of contact formation and breakage to be highly transient, indicative of the local environment within the dense phase showing a dynamic exchange of residue partners. This rapid contact dynamics within the condensates is thought to be the reason behind their ability to facilitate biochemical reactions at a fast rate[66]. Taken together, we concluded that short equilibrium simulations to characterize the contact lifetimes can provide insights into the material properties of charge-rich IDP condensates.

For our model and natural proteins, we found that the interactions in the dense phase were highly similar to that of the interactions for a single-chain. Such a similarity rationalized the observed strong correlation between the different material properties and the single-chain structural properties, as quantified via the radius of gyration, from both the HPS and Martini models, a pivotal discovery of this work. While previous theoretical, computational, and experimental studies have revealed that the coil-to-globule transition of single chains can be used to predict the phase behavior for a wide range of protein sequences, our observation of the correspondence between the single-chain size and the condensate material properties is a distinct result, which could become an invaluable tool for interpreting and predicting condensate dynamics. Our findings indicate that the sequence-level attributes of IDPs can be harnessed to modulate the material properties emergent upon their condensation, allowing for the rational design of synthetic membraneless organelles.

We found that the material properties of the condensates computed from our simulations were quantitatively comparable with experimental measurements of charge-rich protein condensates, when the timescales were rescaled based on the single-chain dynamics (a standard practice in soft matter physics; see "Methods" section). We also demonstrated that our findings were independent of the specific CG models used, by simulating the protein sequences using the Martini model. Moreover, our predictions regarding the dense phase and interfacial properties of IDP condensates can be tested through routine experimental characterizations: the translational diffusion coefficient can be measured through FRAP experiments[66] or through fluorescence correlation spectroscopy (FCS)[16], while the viscosity of the condensates can be obtained using single particle tracking (SPT) microrheology[27] (see Supplementary Tables 2 and 4 for method details). Direct measurements of surface tension can be achieved through the use of dual optical traps (dual-OT) for stretching the condensates[64] (see Supplementary Table 4 for method details). Additionally, it is feasible to deduce the condensate material properties by studying the conformations of an isolated protein chain, which are accessible through fully atomistic simulations or experiments, namely via single-molecule Förster resonance energy transfer (FRET)[76] and small-angle X-ray scattering (SAXS)[77]. We believe that the molecular insights provided in this work will aid in further theoretical developments[78] as well as computational and experimental investigations of diverse sequence features in dictating condensate dynamics, leading to a comprehensive molecular language for the material properties of biomolecular condensates.

## Methods

### Generation of sequences studied

To investigate the effect of charge patterning on the material properties of charged disordered proteins, we employed model polyampholytic sequences comprised of negatively charged glutamic acid (E) residues and positively charged lysine (K) residues. These sequences, which had zero net charge, were selected from a large set of E-K variants generated in our previous study[37], such that they spanned the entire range of nSCD (i.e., from a perfectly alternating sequence with nSCD=0 to a diblock sequence with nSCD=1). We also investigated the charge patterning effects on sequences with a nonzero net charge by simulating the disordered regions of two naturally occurring proteins, namely LAF1 and DDX4[29,32]. Specifically, we took the wild-type sequences of LAF1's RGG domain and DDX4's N-terminal domain, and shuffled the residues within them to generate respective variants with increasing charge segregation. This procedure allowed us to investigate the LAF1 disordered variants with nSCD ranging from 0.003 to 0.394 and the DDX4 disordered variants with nSCD ranging from 0.002 to 0.478. We note that nSCD is strongly correlated with the parameter $\kappa$ (quantified based on the local and global charge asymmetries within an IDP)[22] for all investigated sequences (Supplementary Fig. 14), indicating that our interpretations regarding the condensate material properties and single-chain conformations are not influenced by the choice of the patterning parameter.

### Hydropathy scale (HPS) model

We used our recently developed CG framework to model the IDP sequences as flexible chains with a single bead per residue representation[46,70]. Directly bonded residues interacted with each other via the harmonic potential

$$U_{\mathrm{b}}(r) = \frac{k_{\mathrm{b}}}{2}\left(r - r_0\right)^2,\qquad(5)$$

with distance $r$ between residues, spring constant $k_{\mathrm{b}} = 20\,\mathrm{kcal}/\left(\mathrm{mol\mathring{A}}^2\right)$, and equilibrium bond length $r_0 = 3.8\mathring{A}$. Interactions between nonbonded residues were modeled using a modified Lennard–Jones potential (LJ) that facilitates the attraction between residues $i$ and $j$ to be scaled independently of the short-range repulsion by their average hydropathy $\lambda = \left(\lambda_i + \lambda_j\right)/2$[79,80]:

$$U_{\mathrm{vdW}}(r) = \begin{cases} U_{\mathrm{LJ}}(r) + (1-\lambda)\varepsilon, & r \le 2^{1/6}\sigma \\ \lambda U_{\mathrm{LJ}}(r), & \text{otherwise} \end{cases},\qquad(6)$$

where $U_{\mathrm{LJ}}$ is the standard LJ potential

$$U_{\mathrm{LJ}}(r) = 4\varepsilon\left[\left(\frac{\sigma}{r}\right)^{12} - \left(\frac{\sigma}{r}\right)^6\right].\qquad(7)$$

The parameters of $U_{\mathrm{vdW}}$ include the average diameter $\sigma = \left(\sigma_i + \sigma_j\right)/2$ of residues $i$ and $j$, and the interaction strength $\varepsilon = 0.2\,\mathrm{kcal}/\mathrm{mol}$. For the E-K sequences, the hydropathy $\lambda$ values based on the Kapcha–Rossky scale were used[70,81], while for the LAF1 and DDX4 sequences, the $\lambda$ values were based on the Urry scale[82] (unless otherwise specified), which captures the changes in the phase behavior of natural proteins upon mutations of arginine to lysine and tyrosine to phenylalanine[46,82]. We truncated the pair potential $U_{\mathrm{vdW}}$ and its forces to zero at a distance of $4\sigma$. Further, the nonbonded charged residues interacted via a Coulombic potential with Debye–Hückel electrostatic screening[83]

$$U_{\mathrm{e}}(r) = \frac{q_i q_j}{4\pi\epsilon_{\mathrm{r}}\epsilon_0 r}e^{-r/\ell},\qquad(8)$$

with vacuum permittivity $\epsilon_0$, relative permittivity $\epsilon_{\mathrm{r}} = 80$, and Debye screening length $l = 10\mathring{A}$. The chosen values of $\epsilon_{\mathrm{r}}$ and $l$ correspond to an aqueous solution with a physiological salt concentration of ~100 mM. We truncated the electrostatic potential and its forces to zero at a distance of $35\mathring{A}$.

### Simulation details for the HPS model

For characterizing the translational motion of IDP chains within the dense phase, we simulated the charge-patterned variants in a cubic simulation box at a constant pressure of $P = 0$ atm for a total duration of $0.5\mu s$. The sequence variants attained their preferred dense phase concentration $\rho$ at the end of this simulation run, after which we switched to Langevin dynamics (LD) simulations in the canonical ensemble to simulate for a duration of $1\mu s$. For all the sequences, a damping factor of $t_{\mathrm{damp}} = 1\,\mathrm{ns}$ was used to set the friction coefficient of a residue of mass $m_i$ in the chain to $f_i = m_i/t_{\mathrm{damp}}$. The first $0.2\mu s$ of LD simulations was considered as the equilibration period and the remaining $0.8\mu s$ of the simulation trajectory was used for the computation of the MSD of the residues within a chain in the dense phase, from which the translational diffusion coefficient $D$ was extracted. For sufficiently long times $t \gg \tau_R$, $\tau_R$ being the Rouse relaxation time of the entire chain, the MSD of the inner residues of a chain becomes identical to that of the chain's center of mass[49], which is the case for all investigated E-K, LAF1, and DDX4 sequences (Supplementary Figs. 15a–17a). We further corroborated this behavior by computing the MSD of inner residues relative to the chain's center of mass $g_2$, which plateaued at long times for all sequences (Supplementary Figs. 15b–17b). We also found that the MSD of only the end monomers in a chain were quantitatively the same as the MSD of the chain's inner residues at long times (Supplementary Figs. 15a–17a), indicating that the reported $D$ values are indeed an accurate representation of the translational motion of the IDP chains within the condensates.

We characterized the shear viscosity $\eta$ of the dense phase of IDP condensates through the NEMD simulation technique[51–53]. Specifically, the charge-patterned variants at their preferred concentration $\rho$ were subjected to a steady shear strain in the $x$-direction at different shear rates $\dot{\gamma}$ using the SLLOD equations of motion and the sliding-brick periodic boundary conditions. We observed a linear velocity $v_x$ profile in the gradient direction $y$ and found that its slope, which gives the shear rate, is nearly the same as that of the applied value for all IDP sequences (Supplementary Figs. 18–20). This behavior confirmed that the system is responding as intended at both low and high $\dot{\gamma}$ used in this work. The resulting shear stress $\tau_{xy}$ was then measured as a function of time. We used the $\dot{\gamma}$ and $\tau_{xy}$ values to determine $\eta$ of the dense phase of protein sequences. These simulations were carried out for a total duration of $0.8\,\mu s$ in the case of E-K sequences and $1.2\,\mu s$ in the case of LAF1 and DDX4 sequences. For both model and natural proteins, we considered the trajectories of the first $0.05\mu s$ duration as part of the equilibration period and hence, discarded them in the computation of $\eta$. We also verified that $\eta$ converged to its mean value at times shorter than the simulation duration (Supplementary Fig. 21).

To further verify the convergence of the zero-shear viscosity $\eta_0$, we also computed it from equilibrium MD simulations (for a total duration of $2.5\mu s$) of the E-K sequences using the Green–Kubo relation[65,84]:

$$\eta_0 = \int_0^\infty G(t)dt,\qquad(9)$$

where $G(t)$ is the shear stress relaxation modulus. We measured $G(t)$ based on the autocorrelation of the pressure tensor components $P_{ab}$

(Supplementary Fig. 22a)[85,86],

$$G(t) = \frac{V}{5k_B T}\left[\left\langle P_{xy}(0)P_{xy}(t)\right\rangle + \left\langle P_{xz}(0)P_{xz}(t)\right\rangle + \left\langle P_{yz}(0)P_{yz}(t)\right\rangle\right]$$
$$+ \frac{V}{30k_B T}\left[\left\langle N_{xy}(0)N_{xy}(t)\right\rangle + \left\langle N_{xz}(0)N_{xz}(t)\right\rangle + \left\langle N_{yz}(0)N_{yz}(t)\right\rangle\right],$$

(10)

where $V$ is the volume of the simulation box and $N_{ab} = P_{aa} - P_{bb}$ is the normal stress difference. We computed $\eta_0$ in two different ways: the first approach involved obtaining the measurements by simply integrating $G(t)$ of different E-K sequences (Supplementary Fig. 22b). In the second approach, we followed the method outlined in the recent article by Tejedor et al.[85], where the influence of typical noisy behavior of $G(t)$ at long timescales has been taken care of by fitting the values of $G(t)$ beyond the time after which the intramolecular oscillations have decayed to a series of Maxwell modes ($G_i \exp(-t/\tau_i)$ with $i = 1....4$) equidistant in logarithmic time (Supplementary Fig. 22a)[87]. We then obtained $\eta_0$ as the sum of numerical integration at short times and analytical integration based on the fitted data at long times (Supplementary Fig. 22b). We found that the $\eta_0$ values obtained based on the Green–Kubo relation are in excellent quantitative agreement with those obtained from the NEMD simulations, highlighting that the viscosity values reported in this work accurately represent the rheological properties of the IDP chains in the dense phase.

The surface tension $\gamma$ of the IDP condensates was characterized by performing LD simulations of protein chains in a slab geometry. These simulations were carried out using a damping factor of $t_{damp} = 1$ns to set the residue friction coefficient $f_i$ in the same way as was done in the simulations of protein sequences in a cubic geometry. The protein chains were initially placed in a dense slab within the rectangular simulation boxes ($150\,\text{Å} \times 150\,\text{Å} \times 1200\,\text{Å}$) and were simulated for a duration of $3\mu$s. The last $2.5\mu$s simulation trajectory was used in computing the $\gamma$ values. Again, we ensured that the total simulation time was much longer than the time it took for $\gamma$ to converge to the reported mean value for the investigated IDP sequences (Supplementary Fig. 23).

We also performed the simulations of LAF1 and DDX4 protein chains in a slab geometry at different $T$ for characterizing their phase diagrams. These simulations were carried out at each $T$ for a duration of $0.5\mu$s, and the dense phase and dilute phase concentrations were computed based on the last $0.3\mu$s of the simulation trajectory. Following the previous literature studies[68,70,88], we estimated the critical temperatures by fitting the phase diagram using the law of coexistence densities and the critical densities by assuming the law of rectilinear diameter holds.

To characterize the effect of charge patterning on the single-chain conformations of the charged IDPs, we previously simulated the E-K variants in a cubic box of edge length $160\text{Å}$[37]. We used the same data in this study for establishing its correlations with the material properties of the condensates formed by the E-K variants. In this work, we additionally characterized the single-chain conformations of the LAF1 and DDX4 disordered sequence variants by placing them in a cubic box of edge length $1000\text{Å}$. We chose such large box sizes to prevent unphysical self-interactions between the chain and its periodic images. These simulations were carried out for a duration of $1\mu$s.

Regarding the analysis of contact formation in the dense phase and within a single chain, we computed the probability of a pair of residues $i$ and $j$ in a sequence of length $N$ to be in contact as $P = \langle n_{ij}\rangle$, with $n_{ij} = 1$ if the distance between the pair was less than the cut-off radius of $1.5\sigma$, and $n_{ij} = 0$ if not. This procedure resulted in an $N \times N$ matrix of contact probabilities. Instead of summing up all the contact probabilities $\sum P$ in each column that would result in a one-dimensional vector of length $N$, we resorted to computing $\sum P$ for each residue in the sequence based on the following five

classifications: (1) charged residue $i$ with all oppositely charged residues $j$ in the sequence, (2) charged residue $i$ with all other like charged residues $j$ in the sequence, (3) charged residue $i$ with all uncharged residues $j$ in the sequence, (4) uncharged residue $i$ with all other uncharged residues $j$ in the sequence, and (5) uncharged residue $i$ with all charged residues $j$ in the sequence. As an example, for the LAF1 sequence ($N = 168$) in which there are 44 charged residues and 124 uncharged resides, the above classification resulted in a $\sum P$ vector of length $(3 \times 44) + (2 \times 124) = 380$ for the single-chain and in the dense phase, which we have plotted in Supplementary Fig. 10.

All the physical quantities were averaged over two independent replicas, with each replica divided into three blocks for estimating the standard error of mean. A total of 500, 150, and 106 chains of E-K, LAF1, and DDX4 sequences, respectively, were simulated, with periodic boundary conditions applied to all three Cartesian directions and with a timestep of 10 fs. The equilibrium simulations for characterizing the dense phase dynamical properties, the single-chain simulations for characterizing the IDP conformations, and the phase coexistence simulations for characterizing the surface tension of IDP condensates were carried out using HOOMD-blue (version 2.9.3)[89] with features extended using azplugins (version 0.10.1)[90]. The non-equilibrium shear simulations and equilibrium simulations (using fix ave/correlate/long) for characterizing the dense phase rheological properties were carried out using LAMMPS (29 October 2020 version)[91].

## Connecting simulation and experimental timescales

Using Å, g/mol, and kcal/mol as our units of length, mass, and energy, respectively, the intrinsic simulation time is typically defined as $\tau_{MD} = \sqrt{(\text{g/mol})\text{Å}^2/(\text{kcal/mol})} = 48.89\text{fs}$. However, this choice for $\tau_{MD}$ is usually not suitable for interpreting the time-dependent mechanical properties from CG simulations, as it reflects the thermal fluctuations of the monomer beads, which lack the atomic-level representation of the residues[92]. Instead, we systematically derived experimentally relevant timescales based on the translation diffusive motion of a single protein chain in dilute solution in what follows.

In LD simulations, the Rouse dynamics describes the long-time diffusion coefficient of a single-chain[87]

$$D_{sim} = \frac{k_B T}{N f_i},$$

(11)

where $k_B T$ is the energy scale and $f_i$ is the friction coefficient of monomer $i$ in a chain. In terms of LJ units, we computed $D_{sim} = 27.99\sqrt{\varepsilon\sigma^2/m}$, $9.07\sqrt{\varepsilon\sigma^2/m}$, and $6.45\sqrt{\varepsilon\sigma^2/m}$ for single-chain polymers of lengths corresponding to the E-K ($N = 50$), LAF1 ($N = 168$), and DDX4 ($N = 236$) sequences. These $D_{sim}$ values were obtained for the $f_i$ used in our dense phase simulations.

Next, we estimated the experimentally expected diffusion coefficient of the single-chain IDP sequences in water using the Stokes–Einstein relation[87]

$$D_{exp} = \frac{k_B T}{6\pi\eta_w R_h},$$

(12)

where $\eta_w$ is the viscosity of water and $R_h$ is the hydrodynamic radius of the chain. Given that the E-K and DDX4 sequences were simulated at $T = 300$K, while the LAF1 sequences were simulated at $T = 280$K, we used $\eta_w(T = 300K) = 0.854$mPa $\cdot$ s and $\eta_w(T = 280K) = 1.434$mPa $\cdot$ s, respectively[93], for computing $D_{exp}$. Further, we used $R_h = 16.38\text{Å}$ (most well-mixed E-K that phase separated), $26.56\text{Å}$ (LAF1 wild-type), and $29.69\text{Å}$ (DDX4 wild-type) obtained based on the Kirkwood approximation[94,95] from our simulations. We matched $D_{exp}$ with $D_{sim}$ to get timescales of 1.782ns, 1.683ns, and 0.744ns for the E-K, LAF1, and DDX4 chains respectively. We used these timescales, obtained

based on the mapping at the single-chain level for which the solvent viscosity dominates, to interpret the dense phase dynamical and rheological properties of our IDP sequences. More details on the computations involved in the timescale mapping strategy are given in Supplementary Table 3.

## Experimental measurements of viscosity and surface tension

We state the different experimental methods used by the literature studies[15,64] for measuring the viscosity and surface tension of a wide range of protein condensates (Supplementary Table 4) that we have used for comparison with the corresponding simulation measurements of our IDP sequences in Fig. 3b. All values of the viscosity of condensates were measured directly through experiments such as micropipette aspiration and single particle tracking microrheology. However, 3 out of 11 experimental measurements of surface tension (i.e., for polyK, polyR, and [RGRGG]$_5$ + dT40) were indirectly estimated based on the viscocapillary model, which assumes condensates as a purely viscous medium[16]. Recent studies have shown that the condensates display a time-dependent viscoelastic behavior[14,64,96,97], a finding that questions the use of the viscocapillary model. We and others have also shown that many condensates formed by disordered proteins remain predominantly viscous over the experimentally measured time[24,66]. Thus, we assumed that the surface tension data obtained based on the viscocapillary model can be considered as "estimates" at least for the disordered protein droplets that show predominantly viscous characteristics.

To justify our assumption further, we used the directly measured values of fusion time $\tau_f$, viscosity $\eta$, and surface tension $\gamma$ for four protein condensates formed by oppositely charged binary mixtures (Supplementary Table 5)[64]. Using the values of $\tau_f$ and $\eta$, we estimated the surface tension based on the viscocapillary model $\gamma = 1.97\eta R/\tau_f$ (Supplementary Table 5), where $R$ corresponds to the droplet radius. The factor of 1.97 in the viscocapillary model comes from the observation that the time evolution of the edge-to-edge distance of two fusing viscous droplets, initially with equal radius $R$ (Stokes model), quantitatively followed the stretched exponential functional form, with stretching exponent $\beta = 1.5$ and fusion time $\tau_f = 1.97\frac{\eta R}{\gamma}$ [56,64,98]. We used $R = 3\mu m$ to estimate surface tension, the value for which the fusion times were reported. We found that for the two condensates formed by fully disordered proteins (polyK + heparin and proline-rich motif + heparin), the estimated surface tension was in good agreement with the directly measured surface tension values (a relative difference of <6% when considering the upper bound of the directly measured surface tension value). However, when one of the components consisted of folded domains (SH3 domain+proline-rich motif), the relative difference increased to ~15%, but the estimated value is still in the comparable range with the measured value. The viscocapillary model does poorly when involving condensates made of components that all have folded regions (SH3 domain + lysozyme). Thus, the viscocapillary model seems to be a reasonable choice for estimating the surface tension of condensates formed by fully disordered proteins but not necessarily for those involving folded domains. However, given that the spatiotemporal evolution of condensates is only beginning to be explored, we advise caution in inferring the surface tension of condensates through indirect measurements via the viscocapillary model, particularly when environmental variables such as salt and ATP[99] play a role in regulating the thermodynamics and dynamics of biomolecular condensates.

## Martini model and simulation details

We simulated the E-K sequences using the latest Martini force field version 3, which models the solvent and ion particles explicitly[100]. The protein chains were coarse-grained by applying the martinize2 python script on the atomistic coordinates of extended conformations of E-K sequences. Nonbonded interactions in Martini simulations were modeled using the Verlet cut-off scheme, and a cut-off

distance of 11Å was used for the van der Waals interactions and electrostatic interactions. We treated the long-range electrostatic interactions using the reaction-field method with a dielectric constant of 15[101].

We dispersed 80 chains close to the center of a simulation box, representing a slab geometry of size 100 Å × 100 Å × 740 Å. Following this procedure, the box was solvated with water (~50,000 particles) using the insane python script[102]. We added ions at a concentration of 100 mM (Supplementary Fig. 3a). We energy minimized the systems in a slab geometry by using the steepest descent algorithm for 0.3ns with a timestep of 30fs. Then, we equilibrated the system via NPT simulations at $T = 300$K and $P = 1$bar (applied only in the $z$-direction) for a duration of 20ns (timestep of 20fs). We used the velocity-rescaling thermostat[103] with a time constant of 1ps and Parrinello–Rahman barostat[104] with a time constant of 12ps for maintaining the desired temperature and pressure, respectively. We then carried out the production run for a duration of 18μs at the same conditions as the equilibration run. This duration allowed for the protein and water to reach their equilibrium dense phase concentrations $\rho$, after which we cut out the dense phase section to simulate it in a cuboid simulation box (Supplementary Fig. 3a).

The NVT simulations in a cuboid box were done at a salt concentration of 100 mM to emulate the conditions of our previous CG simulations with the HPS model. Depending on the water concentration within the dense phase of each sequence, the number of water particles in these simulations varied between 9,000 and 14,000. For computing the MSD of the residues of an IDP chain in the dense phase, we carried out these simulations at $T = 300$K for a duration of 10μs. We extracted the values of $D$ for the E-K sequences at long times ($t \geq 5\mu$s) where the relation MSD $= 6Dt$ was observed to hold (Supplementary Fig. 24). For computing the zero-shear viscosity $\eta_0$ based on the Green–Kubo relation[105], we carried out the dense phase simulations for 10 replicas, each for a duration of 2μs. The final $\eta_0$ profile as a function of simulation time was obtained as an average over all replicas (Supplementary Fig. 25). We extracted the values of $\eta_0$ for the E-K sequences from the plateau region at long times ($t \geq 0.5\mu$s). We found that the dynamics in the Martini model was ~2 orders of magnitude slower than that observed in the HPS model, when the respective intrinsic MD timescales were used in both models. For example, the values of $\eta_0$ for the most well-mixed sequence that phase separated (nSCD=0.017) and the diblock sequence (nSCD=1) from the Martini model were 46.23 mPa·s and 291.74 mPa·s, respectively, while those from the HPS model were 0.087 mPa·s and 1.315 mPa·s, respectively. Finally, we also performed 5 replicas of single-chain simulations in the presence of ~30,000 water particles and 100 mM salt concentration in a cubic box of edge length 150Å using the Martini model, each for a duration of 1μs (timestep of 20fs), to compute their $R_g$. All the simulations were carried out using GROMACS (version 2023.1)[106].

## Reporting summary

Further information on research design is available in the Nature Portfolio Reporting Summary linked to this article.

## Data availability

Raw simulation trajectory data used for the analysis to obtain the source data are available from the corresponding authors on request. The starting and ending configurations of the different types of simulations performed using the hydropathy scale and Martini models along with the input scripts to initiate the simulations using the publicly accessible software engines are available via the GitHub repository https://github.com/dsd993/CondensateMaterialProperties_Simulations_Analysis. Source data are provided with this paper.

## Code availability

Analysis codes and the functionalities within the software engines used for computing the physical quantities presented in this work are available via the GitHub repository https://github.com/dsd993/CondensateMaterialProperties_Simulations_Analysis.

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

## Acknowledgements

This research article is based on the work supported by the National Institute of General Medical Science of the National Institutes of Health under the grant R01GM136917 and the Welch Foundation under the grant A-2113-20220331. A.N. acknowledges funding by the Deutsche Forschungsgemeinschaft (DFG, German Research Foundation) through Project 470113688. Y.C.K. is supported by the Office of Naval Research through the U.S. Naval Research Laboratory base program. The computational resources provided by the Texas A&M High Performance Research Computing (HPRC) are gratefully acknowledged.

## Author contributions

D.S.D. and J.M. conceived the research presented in this material. D.S.D designed the sequences and conducted the simulations using the HPS model. D.S.D. and B.S.M. conducted the Martini simulations. D.S.D., A.N., Y.C.K., and J.M. designed the analyses. D.S.D., J.W., B.S.M, and S.R. analyzed the simulation data. D.S.D., A.N., and J.M. wrote and edited the manuscript with critical input from others.

## Competing interests

The authors declare no competing interests.
