## [Peer Review File · Nature Communications]

Reviewers' Comments:

Reviewer #1:

Remarks to the Author:

Devarajan and coworkers use equilibrium and non-equilibrium coarse-grained simulations to study the effects of charge patterning of intrinsically disordered protein sequences on the material properties of the biomolecular condensates they form. They report robust, clear dependence of condensate dynamics on charge separation in proteins, and show evidence of strong correlation between single-chain structural properties and condensate material properties. Considering the importance of biomolecular condensates in general, and the open challenge of understanding and sculpting their material properties in terms of the properties of the underlying protein sequences, the article represents a timely and relevant contribution. However, in order to be considered for publication, the following comments should be adequately addressed.

1. The statements that the authors make depend greatly on the degree of convergence of the underlying simulations. The authors should provide quantitative evidence of convergence of their simulations or give strong arguments as to why absolute convergence may not be relevant for the particular analyses they make.
2. When calculating the diffusion coefficients, the authors monitor the MSD of individual residues and take the average. This approach should be better explained and defended as the translational diffusion coefficients are typically calculated with respect to the center-of-mass motion. In the approach followed by the authors, one could get non-zero diffusion coefficients, even in cases where the center-of-mass does not move at all.
3. In calculating the effective interaction strength from PMFs (p. 14), it is not clear what the first minimum is related to when it comes to the unbound state. In other words, the effective interaction strength should be the difference between the first minimum and the unbound state energy. What did the authors take for the latter?
4. The authors observe a close parallel between single-chain and condensate behavior, regardless of how patterned their sequences are. On the other hand, a particular patterning of stickers and spacers was shown to be critical for the parallel between single-chain and condensate behavior to hold (see, for example, <https://doi.org/10.1038/s41557-021-00840-w> or <https://doi.org/10.1126/science.aaw8653>). The authors should analyze their sequences in light of the omega parameter, developed by Pappu, Mittag and coworkers, and analyze whether indeed patterning and the parallel between single-chain and phase behavior go hand in hand.
5. Why did the authors use different hydrophathy lambda values for E-K sequences as opposed to LAF1 and DDX4 sequences? How different are these?
6. As the study is purely computational, the authors should put more effort into explaining how their results and predictions can be experimentally tested. In light of this, the authors should clearly indicate in the abstract that their study is exclusively computational and also explicitly mention the kind of methods they use. Similarly, the authors should consider making the title more specific and precise.
7. The first two sentences of the Results & Discussion section should be rephrased as they seem to contradict each other. It is not clear what "different sequence characteristics" (mentioned in the second sentence) is, as opposed to sequence characteristics mentioned in the first sentence.

Reviewer #2:

Remarks to the Author:

This paper presents a thorough computational study of the material properties of disordered proteins, based on a coarse-grained (CG) model. The calculations follow well-established methods, are rigorous (within the limit of the CG model) and extensive. There are two major limitations.

1. The first limitation is the CG model. Viscosities predicted by this model are off by 10000-fold from experimental values. Such poor predictions call into questions their relevance to real IDPs. Fig. 3b purport to show better predictions for interfacial tension. However, most of the "experimental" values do not seem to be direct measurements, but rather derived from viscosity and inverse capillary velocity (related to fusion speed), by assuming condensates as viscous. This assumption has now been tested experimentally and found to be erroneous, as condensates are viscoelastic as opposed to purely viscous -- see PMID 34645832. There is now a growing list of reports of experimental results for condensate properties (e.g., refs 13-17 and PMID 34645832). In comparison, calculated results with orders of magnitude errors have limited value.

2. Besides the above limitations of the numerical values, the qualitative findings are largely what is expected from well-known understanding of associative polymers. For example, that long-lived molecular contacts lead to slower dynamics is hardly a surprising observation. Indeed, the basic tenet of associative polymer theory is that contact dynamics is a key determinant of viscoelasticity. The one very interesting result is the similarity in molecular interactions between single chain and condensates.

In short, the model-specific (numerical) results are in serious errors, and the qualitative observations, apart from the one regarding molecular interactions, are mostly expected from associative polymer theory.

Response to Reviews and Summary of Changes Made

Manuscript #: NCOMMS-23-21774

We thank the reviewers for their useful feedback, which has helped us improve the presentation of our results and better highlight their significance. Our responses to the specific issues raised by the reviewers are presented along with the summary of changes made to the manuscript below. The comments from the reviewers are colored in **black**. Our response and the summary of changes made to the manuscript are colored **brown**. Further, the changes made to the article are highlighted in **yellow** in the revised manuscript document.

Reviewers' Comments to Author:

Reviewer #1 (Remarks to the Author):

Devarajan and coworkers use equilibrium and non-equilibrium coarse-grained simulations to study the effects of charge patterning of intrinsically disordered protein sequences on the material properties of the biomolecular condensates they form. They report robust, clear dependence of condensate dynamics on charge separation in proteins, and show evidence of strong correlation between single-chain structural properties and condensate material properties. Considering the importance of biomolecular condensates in general, and the open challenge of understanding and sculpting their material properties in terms of the properties of the underlying protein sequences, the article represents a timely and relevant contribution. However, in order to be considered for publication, the following comments should be adequately addressed.

We thank the reviewer for their encouraging and constructive assessment of our work. Our point-by-point responses to the comments can be found below.

1. The statements that the authors make depend greatly on the degree of convergence of the underlying simulations. The authors should provide quantitative evidence of convergence of their simulations or give strong arguments as to why absolute convergence may not be relevant for the particular analyses they make.

Response: We thank the reviewer for raising this relevant concern regarding the convergence of our reported simulation data. In this work, we characterized various material properties of condensates formed by intrinsically disordered proteins (IDPs) using both

equilibrium and non-equilibrium computational techniques. Next, we provide information that demonstrates the convergence of the material properties reported in the manuscript.

Fig. R1. Running average of surface tension γ as a function of simulation time t for select (a) E–K (nSCD = 0.017), (b) LAF1 (nSCD = 0.263), and (c) DDX4 (nSCD = 0.194) sequences. The solid lines represent the mean value of γ reported in the manuscript for the respective sequences.

We simulated IDPs in a slab geometry¹ to characterize the **surface tension** γ at the interface between the dense and dilute phases. Since the measurement of γ relies on the pressure tensor values that show significant fluctuations (as expected) in the equilibrium molecular dynamics (MD) simulations, we simulated IDPs for a substantially long time ($t = 3 \mu\text{s}$). To provide quantitative evidence for the convergence in the reported surface tension values, we computed the running average of γ as a function of time (**Fig. R1**). We find that the γ estimates fluctuate at short times, but converge to the mean value reported in the manuscript after $t \gtrsim 0.5 \mu\text{s}$ for the E–K, LAF1, and DDX4 sequences. Importantly, the simulation time is much longer than the convergence time.

For characterizing the **shear viscosity** η of the dense phase of IDP condensates, we used non-equilibrium MD simulations,² where we measured the shear stress response τ_{xy} for a specified shear rate $\dot{\gamma}$. Since our objective was to characterize the zero shear viscosity η_0 which is extracted from the Newtonian plateau typically observed at low $\dot{\gamma}$, we ran these simulations for long times ($t = 0.8 \mu\text{s}$ for the E–K sequences, and $t = 1.2 \mu\text{s}$ for the LAF1 and DDX4 sequences). First, we verified that the system exhibits a linear velocity profile in the gradient direction y , in response to the applied low and high $\dot{\gamma}$ (**Figs. R2, S16, and S17**). Further, we find that the actual gradient of velocity in the shearing direction x , which gives the shear rate, is nearly the same as that of the applied shear rate. This behavior confirms that the system is responding as intended both at the low and high shear rates used in this work. Using the measured shear stress τ_{xy} and shear rate $\dot{\gamma}$ values, we computed η of the dense phase of IDP condensates using Newton’s law of viscosity.³

Fig. R2. Velocity v_x of the residues in the flow direction x as a function of gradient direction y in the dense phase of a select E–K sequence with $nSCD = 0.468$ at (a) low and (b) high shear rates used in this work. The solid line corresponds to a linear fit to the data, the slope of which gives the shear rate $\dot{\gamma}_{fit}$. The value of $\dot{\gamma}_{fit}$ is nearly identical to the imposed shear rate $\dot{\gamma}_{input}$ on the dense phase of the condensate.

To establish convergence in the reported η values of the dense phase of IDP condensates, we computed its running average as a function of time (**Fig. R3**). Much like the surface tension, we find that the η values converge to the mean value at times shorter than the total simulation time for the investigated IDP sequences. These observations indicate that the different material properties reported in this work accurately reflect the interfacial and dense phase characteristics of IDP condensates.

Fig. R3. Running average of viscosity η as a function of simulation time t at a low shear rate for select (a) E–K ($nSCD = 0.468$), (b) LAF1 ($nSCD = 0.010$), and (c) DDX4 ($nSCD = 0.021$) sequences. The solid lines represent the mean value of η for the low shear rate reported in the manuscript for the respective sequences.

Regarding the reliability of the reported **diffusion coefficient D** of protein chains in the dense phase obtained from the equilibrium simulations, we request the reviewer to see our response to the next comment.

Changes made to the manuscript: We have included the figures discussed in this response as new supplementary figures (**Figs. S16-S20**) and the associated discussion highlighting the convergence of our simulations in the Methods section (**Pages 26 and 27**) of the revised manuscript.

2. When calculating the diffusion coefficients, the authors monitor the MSD of individual residues and take the average. This approach should be better explained and defended as the translational diffusion coefficients are typically calculated with respect to the center-of-mass motion. In the approach followed by the authors, one could get non-zero diffusion coefficients, even in cases where the center-of-mass does not move at all.

Fig. R4. (a) Mean square displacement $MSD(t)$ of the residues of a chain (solid lines) and the center of mass of a chain (dashed lines) in the dense phase for select E–K sequences. The slope values 2, 0.5, and 1 indicate ballistic, sub-diffusive, and diffusive regimes, respectively. (b) MSD of the residues of a chain relative to the center of mass $g_2(t)$ in the dense phase for select E–K sequences. To avoid the chain end effects, 20 residues on either end of a chain have not been considered for computing the MSD and g_2 of the individual residues. The line color, ranging from blue to red, indicates increasing nSCD.

Response: We would like to clarify this misunderstanding about the appropriateness of our method to calculate diffusion coefficients D . We computed the mean square displacement (MSD) of the individual residues within IDPs, from which the diffusion coefficient was extracted from the diffusive regime ($MSD \propto t$) as $D = MSD/6t$, where t is the simulation time. For sufficiently long times $t \gg \tau_R$, τ_R being the Rouse relaxation time of the entire chain, the MSD of the residues becomes identical to the MSD of the chain.⁴ In agreement with this expectation, we observe that the MSD of the residues is identical to that of the chain’s center of mass in the diffusive regime

for the E–K, LAF1, and DDX4 sequences (Figs. R4a, S13a, and S14a). In fact, by computing the MSD of individual residues relative to the chain’s center of mass g_2 (Figs. R4b, S13b, and S14b), we find that g_2 plateaus at long times, thus providing quantitative evidence that our reported D values are an accurate representation of the translational motion of the IDP chains.

Changes made to the manuscript: We have included new supplementary figures (Figs. S13-S15) showing the comparison between MSD of individual residues and MSD of chain’s center of mass for select E–K, LAF1, and DDX4 sequences. We have also added a discussion, which justifies the use of the MSD of individual residues to extract the translational diffusion coefficient of the protein chains at long times in the Methods section (Pages 25 and 26) of the revised manuscript.

3. In calculating the effective interaction strength from PMFs (p. 14), it is not clear what the first minimum is related to when it comes to the unbound state. In other words, the effective interaction strength should be the difference between the first minimum and the unbound state energy. What did the authors take for the latter?

Fig. R5. (a) Potential of mean force (PMF) as a function of distance r between the interchain oppositely charged residues for the E–K sequences in the dense phase. (b) Free energy change ΔF between bound and unbound states for the interchain oppositely charged residues as a function of nSCD for the E–K, LAF1, and DDX4 sequences.

Response: We agree with the reviewer that our definition of the effective interaction strength ξ obtained from the potential of mean force (PMF) was not clearly stated. The values of ξ (Fig. 4a) were taken as the absolute difference in the PMF value at the first minimum and the PMF value at the maximum, which follows the first minimum (Fig. R5a). However, upon close examination, we realized that the PMFs of sequences for which nSCD < ~0.20 do not exhibit an attractive well, as opposed to nSCD < 0.02, which we claimed in the manuscript. To correct this

mistake in our previous version, we took the reviewer's suggestion and computed the free energy change ΔF as the difference in PMF value at the first minimum (corresponding to the bound state) and PMF = 0 (corresponding to the unbound state) (**Fig. R5b**). We note that this change does not alter any of the conclusions made in this work.

Changes made to the manuscript: We replaced **Fig. 4a** with the ΔF values in the revised manuscript. We have clearly stated how ΔF is computed and added sentences interpreting **Fig. 4a** in the Results section (**Page 14**) of the revised manuscript.

4. The authors observe a close parallel between single-chain and condensate behavior, regardless of how patterned their sequences are. On the other hand, a particular patterning of stickers and spacers was shown to be critical for the parallel between single-chain and condensate behavior to hold (see, for example, <https://doi.org/10.1038/s41557-021-00840-w> or <https://doi.org/10.1126/science.aaw8653>). The authors should analyze their sequences in light of the omega parameter, developed by Pappu, Mittag and coworkers, and analyze whether indeed patterning and the parallel between single-chain and phase behavior go hand in hand.

Response: We thank the reviewer for bringing up this comment as it points to one of the key findings that similar molecular interactions exist at the single-chain level and within the IDP condensates (**Figs. S7-S9**), which reviewer 2 also highlights. First, we want to clarify that the dependence of sequence patterning for the parallel between single-chain and condensate behavior has been demonstrated for relating protein conformations with only the thermodynamic properties in the articles^{5, 6} mentioned by the reviewer. As per homopolymer-based solution theories,^{7, 8} intrachain contacts translate to interchain contacts in concentrated solutions. Congruently, simulations revealed a strong correlation between the coil-to-globule transition temperature T_θ of a single-chain and the critical temperature T_c of the thermodynamic phase, in the limit of infinitely long homopolymer chains.⁹ Our group¹⁰ first established that T_θ and T_c are strongly correlated for a diverse set of heteropolymeric naturally occurring IDPs with different lengths and compositions, which led to inferring the phase separation propensity of condensates (measured through T_c or saturation concentration c_{sat}) from simple single-chain size measurements (measured through radius of gyration R_g).^{5, 6}

Further, the Ω parameter that the reviewer suggests to use for the analysis of our results was proposed for quantifying the extent of segregation of positively charged (Ω_+) or the extent of segregation of negatively charged residues (Ω_-) with respect to all other residues.¹¹ Given that the LAF1 and DDX4 sequences also contain uncharged residues and that our interest is to specifically quantify the extent of segregation between the oppositely charged residues, the appropriate patterning parameter from Pappu and coworkers would be δ_{+-} (corresponding to the mean squared deviation between the local and global charge asymmetry).¹¹ When δ_{+-} is normalized by the corresponding maximum value achievable for a given composition, we get the κ parameter introduced by Das and Pappu.¹² We find that our nSCD parameter and κ are strongly correlated

Fig. R6. Correlation between nSCD parameter used in this work and κ parameter introduced by Das and Pappu.¹² The solid line corresponds to a linear fit with correlation coefficient $R^2 = 0.90$.

for all investigated sequences (**Fig. R6**), which indicates that our interpretations of the condensate material properties and single-chain conformations would be the same irrespective of the choice of the charge patterning parameter. Moreover, we agree with the reviewer that Pappu, Mittag, and coworkers demonstrated the breakdown in the correlation between single-chain dimension and phase separation propensity (1) when the hydrophobic residues are clustered in the sequence and (2) when the net charge per residue of the sequence keeps deviating from zero,^{5, 6} which we had already stated in the original version of the manuscript. However, such deviations are not observed for the E–K sequences, which is already part of the published literature.^{10, 13}

Changes made to the manuscript: We have rephrased/added sentences in the Results section (**Page 19**) of the revised manuscript for (1) clearly indicating that correlations established in the literature pertain to the phase behavior of condensates, (2) discussing the deviations in the parallel between the single-chain properties and condensed phase thermodynamic properties with the appropriate references that the reviewer has indicated, and (3) stating that such deviations have not been observed with changes in the sequence charge patterning. We have also added a new supplementary figure (**Fig. S12**) and noted the strong correlation between nSCD and κ in the Methods section (**Page 24**) of the revised manuscript.

5. Why did the authors use different hydropathy lambda values for E-K sequences as opposed to LAF1 and DDX4 sequences? How different are these?

Response: We used the Urry hydropathy scale for LAF1 and DDX4 because of its ability to accurately describe the single-chain properties and condensed phase behavior of natural proteins, which is a significant improvement over the originally proposed Kapcha-Rossky (KR)

scale.¹⁴ But we still used the KR hydrophathy scale¹ for the E–K sequences to be consistent with our previous work¹⁵ with this system and because the behavior of these sequences is largely dominated by electrostatic interactions. In the KR scale, the hydrophathy values for the E and K residues are 0.460 and 0.514, whereas their values in the Urry scale are -0.08 and 0.382 . This may appear to be a big difference, but the total pairwise energy profiles are very similar, after taking into account the electrostatic contributions part (**Fig. R7**).

Fig. R7. Potential energy curves (van der Waals vdW + electrostatic interactions) for (a) oppositely charged residues and (b) like charged residues in E–K sequences.

We also directly tested whether our conclusions hold if instead the KR scale was used for the natural proteins by simulating three sequences of LAF1 ($n\text{SCD} = 0.010, 0.131, \text{ and } 0.394$) and DDX4 ($n\text{SCD} = 0.021, 0.194, \text{ and } 0.478$). We find that the strong correlation between the normalized material properties and normalized single-chain size persisted with the inclusion of the values based on the KR scale, with R^2 ranging from 0.84 to 0.91 (**Fig. R8**). These observations highlight that our conclusions are not affected by the model choice.

Changes made to the manuscript: Since the Kapcha-Rosky scale does not describe the natural proteins accurately (for the reasons mentioned in our response), we chose to add **Fig. R8** as a supplementary figure (**Fig. S11**). We referenced the figure and stated that our results are general irrespective of the hydrophathy scales used in the Results section (**Page 20**) of the revised manuscript.

6. As the study is purely computational, the authors should put more effort into explaining how their results and predictions can be experimentally tested. In light of this, the authors should clearly indicate in the abstract that their study is exclusively computational and also explicitly mention the kind of methods they use. Similarly, the authors should consider making the title more specific and precise.

Fig. R8. Correlation (a) between diffusivity D^* and single-chain R_g^* and (b) between surface tension γ^* and single-chain R_g^* for all sequences. The symbol color, ranging from blue to red, indicates increasing nSCD. The values of D^* , γ^* , and R_g^* are obtained after normalizing D , γ , and R_g by those computed for the same reference sequences used in **Figs. 2 and 3**. The closed and open symbols correspond to the values obtained based on the Urry's hydropathy scale and Kapcha-Rosky's hydropathy scale, respectively. The solid lines in correspond to the linear fits $\ln(D^*) = 3.91\ln(R_g^*) - 0.08$ with correlation coefficient $R^2 = 0.84$, and $\ln(\gamma^*) = -4.33\ln(R_g^*) - 0.03$ with $R^2 = 0.91$, respectively.

Response: We have made the changes, implementing all the suggestions of the reviewer, in the revised manuscript.

Changes made to the manuscript: We have added sentences in the Discussion section (**Pages 22 and 23**) of the revised manuscript describing the different experimental techniques that can be leveraged to characterize the different physical properties and to test the correlations reported in this work. We have rephrased the Abstract stating the different techniques used to carry out our purely computational work. In addition, we have changed the title of the manuscript to 'Sequence-dependent material properties of biomolecular condensates and their relation to dilute phase conformations'.

7. The first two sentences of the Results & Discussion section should be rephrased as they seem to contradict each other. It is not clear what "different sequence characteristics" (mentioned in the second sentence) is, as opposed to sequence characteristics mentioned in the first sentence.

Response: We have now rephrased the indicated sentences for better clarity.

Changes made to the manuscript: We rephrased the sentences (**Page 5**) in the revised article as follows: “The effect of charge patterning on the thermodynamic phase behavior of disordered proteins with varying sequence characteristics such as the chain length (N), the fraction of charged residues (FCR), and the net charge per residue (NCPR) is a well-studied problem. For example, Lin and Chan¹³ demonstrated that charge patterning in polyampholytic disordered proteins with fixed N altered their critical temperature T_c for phase separation such that its changes were synonymous with that of their single-chain size. Later, Dignon *et al.*¹⁰ showed that T_c and the single-chain coil-to-globule transition temperature T_θ are strongly correlated for a diverse set of natural disordered proteins, which included sequences with different charge patterning. However, the effect of charge patterning on the condensate material properties (*e.g.*, viscosity and surface tension) of proteins with different sequence lengths and charge content is not well understood.”

Reviewer #2 (Remarks to the Author):

This paper presents a thorough computational study of the material properties of disordered proteins, based on a coarse-grained (CG) model. The calculations follow well-established methods, are rigorous (within the limit of the CG model) and extensive. There are two major

limitations

We thank the reviewer for their constructive feedback on our work, which we believe has provided an opportunity to discuss the predictive capability of our coarse-grained (CG) model. Our point-by-point responses to the comments can be found below. Since the 1st comment of the reviewer raises multiple questions, we have split our response to the comment into two parts below.

1a. The first limitation is the CG model. Viscosities predicted by this model are off by 10,000-fold from experimental values. Such poor predictions call into questions their relevance to real IDPs.

Response: The 10,000-fold discrepancy between simulations and experiments stems from the use of the intrinsic MD timescale $\tau_{\text{MD}} = \sqrt{(\text{g/mol}) \text{ \AA}^2 / (\text{kcal/mol})} = 48.89 \text{ fs}$ and is not a limitation specific to our CG model.¹⁶ The value of τ_{MD} is usually not a good choice for CG simulations, as it is based on the thermal fluctuations of the monomer beads, which lack the atomic-level representation of the residues. In fact, to the best of our knowledge, none of the CG models in the current literature can predict viscosities η_0 that are in quantitative agreement with the experiments if such an intrinsic MD timescale is used. We have already stated this in the previous version of the manuscript: “CG models are known to significantly underpredict viscosities as compared to experimental measurements,¹⁷ which is primarily due to the smoothening of the free energy landscape¹⁸ and inaccurate solvent frictional effects.” It is because of these reasons, we

reported the reduced viscosities (simulation η_0 reduced by the solvent viscosity estimated based on Stokes drag and experimental η_0 reduced by the viscosity of water).

However, we understand that this procedure has caused confusion, which led to the reviewer’s concern about the CG model. We regret not making it clear enough in the manuscript and address this aspect systematically in what follows. Specifically, we describe how the mesoscopic quantity such as the translational diffusive motion of a single protein chain in dilute solution can be used to derive experimental-relevant timescales to interpret the dynamical and rheological properties of the condensed phase obtained from implicit-solvent CG simulations.¹⁹⁻²¹

In Langevin dynamics (LD) simulations, the Rouse dynamics describes the long-time diffusion coefficient of a single-chain²²

$$D_{\text{sim}} = \frac{k_{\text{B}}T}{Nf_i},$$

where $k_{\text{B}}T$ is the energy scale and f_i is the friction coefficient of monomer i in a chain. In terms of Lennard Jones units, we computed $D_{\text{sim}} = 27.99 \sqrt{\varepsilon\sigma^2/m}$, $9.07 \sqrt{\varepsilon\sigma^2/m}$, and $6.45 \sqrt{\varepsilon\sigma^2/m}$ for single-chain polymers of lengths corresponding to the E–K ($N = 50$), LAF1 ($N = 168$), and DDX4 ($N = 236$) sequences. These D_{sim} values were obtained for the f_i used in our dense phase simulations.

To make meaningful comparisons between simulations and experiments, it is necessary to establish the connection between their relevant timescales. We estimated the experimentally expected diffusion coefficient of the single-chain IDP sequences in water using the Stokes-Einstein relation²²

$$D_{\text{exp}} = \frac{k_{\text{B}}T}{6\pi\eta_{\text{w}}R_{\text{h}}},$$

where η_{w} is the viscosity of water and R_{h} is the hydrodynamic radius of the chain. Given that the E–K and DDX4 sequences were simulated at $T = 300$ K, while the LAF1 sequences were simulated at $T = 280$ K, we used $\eta_{\text{w}}(T = 300 \text{ K}) = 0.854 \text{ mPa}\cdot\text{s}$ and $\eta_{\text{w}}(T = 280 \text{ K}) = 1.434 \text{ mPa}\cdot\text{s}$, respectively,²³ for computing D_{exp} . Further, we used $R_{\text{h}} = 16.38 \text{ \AA}$ (well-mixed E–K), 26.56 \AA (LAF1 wild-type), and 29.69 \AA (DDX4 wild-type), which we obtained based on the Kirkwood approximation^{24, 25} from our simulations. We matched D_{exp} with D_{sim} to get timescales of $1.782 \times 10^{-9} \text{ s}$, $1.683 \times 10^{-9} \text{ s}$, and $0.744 \times 10^{-9} \text{ s}$ for the E–K, LAF1, and DDX4, sequences respectively. We used these timescales, obtained based on the mapping at the single-chain level for which the solvent viscosity dominates, to interpret the dense phase dynamical properties of our IDP sequences.

We find that the diffusion coefficients obtained from the dense-phase simulations of the LAF1 and DDX4 wild-type sequences, based on the timescales mapped at the single-chain level,

Disordered sequence	D ($\mu\text{m}^2/\text{s}$) from experiments*	D ($\mu\text{m}^2/\text{s}$) from simulations#
LAF1 wild-type	0.025 ± 0.009	0.052 ± 0.002
DDX4 wild-type	0.400 ± 0.100	0.061 ± 0.003

Table R1. Comparison of dense-phase diffusion coefficients D between simulations (obtained based on the timescales mapped at the single-chain level at infinite dilution) and FRAP experiments for the LAF1 and DDX4 wild-type sequences. *The experimental conditions pertaining to the D values for LAF1 and DDX4 were $T = 289.15$ K to 291.15 K and $T = 293.15$ K, both in 150 mM NaCl buffer. #The simulation conditions pertaining to the D values for LAF1 and DDX4 were $T = 280$ K and $T = 300$ K with Debye screening length $\ell = 10$ Å, which corresponds to 100 mM salt concentration.

were in the comparable range to those obtained from FRAP experiments on the corresponding *in vitro* droplets^{26, 27} (Table R1). Further, a good quantitative agreement in η_0 for the LAF1 WT from simulations and experiments is observed (Fig. R9). In fact, we find that the η_0 obtained for the simulated charge-rich IDP condensates are in close proximity to those obtained for other charge-rich protein condensates, namely PGL3, polyR, polyK, polyK+heparin, SH3-targeting proline-rich motif+heparin and [RGRGG]₅ + dT40 (Fig. R9). These observations highlight the ability of our CG model to capture the material properties of disordered protein condensates when appropriate timescales are used. Note that we reached the same conclusions based on the rescaling we did in the originally submitted manuscript, but the modified approach is simpler to comprehend.

Fig. R9. Comparison of η_0 and γ obtained for the simulated E–K, LAF1, and DDX4 dense phases with those measured for a wide range of protein condensates through experiments.

Changes made to the manuscript: We replaced Fig. 3b with Fig. R9 in the revised manuscript, which also includes the experimental measurement of four other protein condensates

from the article²⁸ mentioned by the reviewer in the next comment. We also added **Tables R1** in the supplementary material (**Table S1**). Instead of the previous discussion on how we obtained the reduced viscosity based on the solvent viscosities, we now discuss the approach outlined in our response in the Results section (**Pages 12-14**) as well as in the Methods section (**Pages 28 and 29**) of the revised manuscript.

1b. Fig. 3b purport to show better predictions for interfacial tension. However, most of the "experimental" values do not seem to be direct measurements, but rather derived from viscosity and inverse capillary velocity (related to fusion speed), by assuming condensates as viscous. This assumption has now been tested experimentally and found to be erroneous, as condensates are viscoelastic as opposed to purely viscous -- see PMID 34645832. There is now a growing list of reports of experimental results for condensate properties (e.g., refs 13 – 17 and PMID 34645832). In comparison, calculated results with orders of magnitude errors have limited value.

Protein condensate	Protein molecular structure	Viscosity (Pa.s)	Surface tension (mN/m)	Brief method details to get surface tension
LAF1 RGG	Fully disordered	1.62 ± 0.18 Micropipette aspiration	0.159 ± 0.010 Micropipette aspiration	From the intercept of the measured shear rate at different aspiration pressure
polyK	Fully disordered	0.204 SPT	0.017 Fusion	Independent measure of viscosity is used in the surface tension to viscosity ratio obtained based on condensates fusion timescale
polyR	Fully disordered	14.4 SPT	0.1 Fusion	Same as polyK
[RGRGG] ₅ + dT40	Fully disordered	3 SPT	0.8 Fusion	Same as polyK
polyK (pK) + heparin (H)	Fully disordered	0.30 ± 0.03 Oscillatory microrheology	0.0571 ± 0.0038 Dual optical traps	Dual optical traps to stretch the condensates and the resulting static spring constant is proportional to surface tension
Proline-rich motif (P) + heparin (H)	Fully disordered	0.53 ± 0.04 Oscillatory microrheology	0.067 ± 0.0085 Dual optical traps	Same as pK + H

Protein condensate	Protein molecular structure	Viscosity (Pa.s)	Surface tension (mN/m)	Brief method details to get surface tension
PGL3	Disordered + folded	1 Dual optical traps	$\sim 4.5 \times 10^{-3}$ Dual optical traps	Dual optical traps to periodically stretch the condensates. Dynamic spring constant and complex shear modulus gives surface tension
FUS	Disordered + folded	0.7 Dual optical traps	$\sim 3.1 \times 10^{-3}$ Dual optical traps	Same as PGL3
NPM1	Disordered + folded	in vitro : 0.74 SPT in vivo : 37 Fusion	in vitro : 8×10^{-4} Sessile drop in vivo : 4×10^{-4} Sessile drop	Fusion: Same as polyK Sessile drop: Condensates of various sizes were imaged through prism. Density, length scales from droplet shape, and gravity were used to obtain surface tension
FIB1	Disordered + folded	~ 100 SPT	1.23×10^{-3} Fusion	Fusion: Same as polyK Sessile drop: Same as NPM1
PGL1	Disordered + folded	~ 1 FRAP	$\sim 4.5 \times 10^{-3}$ Fusion	Same as polyK
LAF1	Disordered + folded	23.4 SPT	0.19 Fusion	Same as polyK
SH3 domain (S) + proline-rich motif (P)	S: disordered + folded P: fully disordered	3.75 ± 0.14 Oscillatory microrheology	0.0734 \pm 0.0058 Dual optical traps	Same as pK + H
SH3 domain (S) + lysozyme (L)	S: disordered + folded P: fully disordered	10.1 ± 1.1 Oscillatory microrheology	0.106 ± 0.015 Dual optical traps	Same as pK + H

Table R2. Surface tension and viscosity of protein condensates and the experimental methods used to measure them in the literature. SPT corresponds to single particle tracking.

Response: We thank the reviewer for raising this important issue regarding the interfacial properties of protein condensates. Our objective is to test if a simple CG model can give us values in the expected range of other biomolecular condensates, which were previously measured or estimated through experiments that are not easy to perform. This work demonstrates the ability of our physics-based CG model to capture the salient material properties of condensates in a sequence-dependent manner.

To address this comment on interfacial properties, we explicitly state the different methods used by the literature studies for the experimental measurements of viscosity and surface tension that we originally reported alongside our corresponding simulation measurements (**Table R2**). In addition, we have included the experimental data for four other protein condensates reported in the article²⁸ that the reviewer has cited in the comment. Out of 11 experimental measurements of surface tension reported in **Fig. 3b** of the originally submitted manuscript, 5 of them were estimated based on the viscocapillary model. We agree with the reviewer that the condensates display a time-dependent viscoelastic behavior²⁸⁻³¹ (characterized by the elastic modulus G' and viscous modulus G''), a finding that questions the use of the viscocapillary model, which assumes condensates as a viscous medium. We and others have recently shown that many condensates formed by disordered proteins remain predominantly viscous over the experimentally measured time.^{32, 33} Thus, one could assume that the surface tension data obtained based on the viscocapillary model are reliable “estimates” at least for the disordered protein droplets that show predominantly viscous characteristics.

Protein condensate	Fusion time (ms)	Viscosity (Pa.s)	Surface tension (mN/m)	Surface tension from viscocapillary model (mN/m)
polyK (pK) + heparin (H)	30.3 ± 0.9	0.30 ± 0.03	0.0571 ± 0.0038	0.0585
Proline-rich motif (P) + heparin (H)	39.3 ± 3.9	0.53 ± 0.04	0.067 ± 0.0085	0.0797
SH3 domain (S) + proline-rich motif (P)	384 ± 31	3.75 ± 0.14	0.0734 ± 0.0058	0.0577
SH3 domain (S) + lysozyme (L)	1774 ± 128	10.1 ± 1.1	0.106 ± 0.015	0.0337

Table R3. Comparison of estimated surface tension based on the viscocapillary model with the directly measured values for four protein condensates formed by oppositely charged binary mixtures.

But to justify our assumption further, we used the directly measured values of fusion time τ_f , viscosity η , and surface tension γ for four protein condensates formed by oppositely charged binary mixtures (**Table R3**) reported in the article²⁸ that the reviewer has brought to our attention.

Using the values of τ_f and η , we estimated the surface tension based on the viscocapillary model $\gamma = 1.97\eta R/\tau_f$ (**Table R3**), where R corresponds to the droplet radius. We used $R = 3 \mu\text{m}$ to estimate surface tension, the value for which the fusion times were reported. We find that for the two condensates formed by fully-disordered proteins (polyK+heparin and proline-rich motif+heparin), the surface tension estimate based on the viscocapillary model is in good agreement with the directly measured surface tension values (the relative difference is $< 6\%$ when considering the upper bound of the directly measured surface tension value). However, when one of the components consists of folded domains (SH3 domain+proline-rich motif), the relative difference increases to $\sim 15\%$ but the estimated value is still in the comparable range with the measured value. The viscocapillary model does poorly when involving condensates made of components that all have folded regions (SH3 domain+lysozyme). Thus, the viscocapillary model seems to be a reasonable choice for estimating the surface tension of condensates formed by fully disordered proteins but not necessarily for those involving folded domains. Based on these observations, we chose to retain the experimental data for the three disordered protein condensates (*i.e.*, polyK, polyR, and [RGRGG]₅ + dT40) that used the viscocapillary model, but removed the data for four protein condensates involving folded domains (*i.e.*, full length LAF1, PGL1, FIB1, and *in vivo* measurement of NPM1) in **Figs. R9 and 3b**.

Changes made to the manuscript: We have added **Tables R2 and R3** in the supplementary material (**Tables S2 and S3**). We have added sentences indicating that some of the surface tension measurements were based on the viscocapillary model along with the concerns regarding its use with an appropriate reference²⁸ in the Methods sections (**Pages 29 and 30**) of the revised manuscript. In addition, we have stated in the same section (**Page 30**) that surface tension estimates based on the viscocapillary model for three disordered protein condensates seem to be a reasonable choice as per **Table S3**. In light of our observation that the viscocapillary model does poorly when folded domains are involved (**Table S3**), we have removed the experimental data points of four protein condensates involving folded domains (*i.e.*, full length LAF1, PGL1, FIB1, and *in vivo* measurement of NPM1) from **Fig. 3b**.

2. Besides the above limitations of the numerical values, the qualitative findings are largely what is expected from well-known understanding of associative polymers. For example, that long-lived molecular contacts lead to slower dynamics is hardly a surprising observation. Indeed, the basic tenet of associative polymer theory is that contact dynamics is a key determinant of viscoelasticity. The one very interesting result is the similarity in molecular interactions between single chain and condensates.

Response: We agree with the reviewer that the observation of long-lived contacts leading to slow condensate dynamics is reminiscent of associative polymer solutions. We have already mentioned in the Discussion section that this observation is in line with polymer theories without explicitly using the term ‘associative polymers’. To rectify this, we have now added a sentence

highlighting that our observation is in line with associative polymer theory with relevant references.³⁴⁻³⁶ However, to the best of our knowledge, it has not been quantitatively shown that the values of different condensate material properties (in the dense-phase as well as at the interface) and contact lifetime would fall on a master curve, exhibiting a strong correlation, for disordered proteins with different sequence characteristics such as the ones investigated in this work. This observation leads to the possibility of using short equilibrium simulations for characterizing the contact dynamics to infer the IDP condensate material properties.

More importantly, the key finding of our work is that the similar molecular interactions observed at the single-chain level and within the condensates of IDPs also lead to strong correlations between the single-chain size and their various condensate material properties. Previously, such a correlation between the single-chain size and phase separation propensity was established by us and others,^{6, 10, 13} which led to numerous studies investigating this aspect for diverse protein sequence features (see our response to the 4th comment of reviewer 1). It is now widely accepted that single-chain simulations of coil-to-globule transitions are powerful tools to interpret the phase behavior of protein sequences. We believe that the correspondence between the single-chain size and the condensate material properties shown in this work would prove to be an invaluable tool for interpreting condensate dynamics.

Changes made to the manuscript: We have rephrased the sentences in the Discussion section (**Page 22**) of the revised manuscript to indicate that stronger interchain interactions leading to slower condensate dynamics is reminiscent of the associative polymer solutions. Similarly, we have clearly stated in the Discussion section (**Page 22**) that the key finding of our work is the similar molecular interactions between the single-chain and condensed phase, which leads to a strong correlation between the different condensate material properties and the single-chain structural properties.

In short, the model-specific (numerical) results are in serious errors, and the qualitative observations, apart from the one regarding molecular interactions, are mostly expected from associative polymer theory.

Response: We hope that our responses above convinces the reviewer about the merits of our work, making it suitable for the readers of Nature Communications.

Changes made to the manuscript: No changes made to the manuscript.

REFERENCES

(1) Dignon, G. L.; Zheng, W.; Kim, Y. C.; Best, R. B.; Mittal, J. Sequence determinants of protein phase behavior from a coarse-grained model. *PLOS Computational Biology* **2018**, *14* (1), e1005941.

- (2) Evans, D.; Morriss, G. Non-Equilibrium Statistical Mechanics of Liquids. Cambridge University Press, Cambridge: 2008.
- (3) Sundaravadivelu Devarajan, D.; Nourian, P.; McKenna, G. B.; Khare, R. Molecular simulation of nanocolloid rheology: Viscosity, viscoelasticity, and time-concentration superposition. *Journal of Rheology* **2020**, *64* (3), 529-543.
- (4) Nikoubashman, A.; Howard, M. P. Equilibrium Dynamics and Shear Rheology of Semiflexible Polymers in Solution. *Macromolecules* **2017**, *50* (20), 8279-8289.
- (5) Martin, E. W.; Holehouse, A. S.; Peran, I.; Farag, M.; Incicco, J. J.; Bremer, A.; Grace, C. R.; Soranno, A.; Pappu, R. V.; Mittag, T. Valence and patterning of aromatic residues determine the phase behavior of prion-like domains. *Science* **2020**, *367* (6478), 694-699.
- (6) Bremer, A.; Farag, M.; Borchers, W. M.; Peran, I.; Martin, E. W.; Pappu, R. V.; Mittag, T. Deciphering how naturally occurring sequence features impact the phase behaviours of disordered prion-like domains. *Nature Chemistry* **2022**, *14* (2), 196-207.
- (7) Flory, P. J. Thermodynamics of High Polymer Solutions. *The Journal of Chemical Physics* **2004**, *10* (1), 51-61.
- (8) Huggins, M. L. Some Properties of Solutions of Long-chain Compounds. *The Journal of Physical Chemistry* **1942**, *46* (1), 151-158.
- (9) Panagiotopoulos, A. Z.; Wong, V.; Floriano, M. A. Phase Equilibria of Lattice Polymers from Histogram Reweighting Monte Carlo Simulations. *Macromolecules* **1998**, *31* (3), 912-918.
- (10) Dignon, G. L.; Zheng, W.; Best, R. B.; Kim, Y. C.; Mittal, J. Relation between single-molecule properties and phase behavior of intrinsically disordered proteins. *Proceedings of the National Academy of Sciences* **2018**, *115* (40), 9929-9934.
- (11) Cohan, M. C.; Shinn, M. K.; Lalmansingh, J. M.; Pappu, R. V. Uncovering Non-random Binary Patterns Within Sequences of Intrinsically Disordered Proteins. *Journal of Molecular Biology* **2022**, *434* (2), 167373.
- (12) Das, R. K.; Pappu, R. V. Conformations of intrinsically disordered proteins are influenced by linear sequence distributions of oppositely charged residues. *Proceedings of the National Academy of Sciences* **2013**, *110* (33), 13392-13397.
- (13) Lin, Y.-H.; Chan, H. S. Phase Separation and Single-Chain Compactness of Charged Disordered Proteins Are Strongly Correlated. *Biophysical Journal* **2017**, *112* (10), 2043-2046.
- (14) Regy, R. M.; Thompson, J.; Kim, Y. C.; Mittal, J. Improved coarse-grained model for studying sequence dependent phase separation of disordered proteins. *Protein Science* **2021**, *30* (7), 1371-1379.
- (15) Sundaravadivelu Devarajan, D.; Rekhi, S.; Nikoubashman, A.; Kim, Y. C.; Howard, M. P.; Mittal, J. Effect of Charge Distribution on the Dynamics of Polyampholytic Disordered Proteins. *Macromolecules* **2022**, *55* (20), 8987-8997.
- (16) Padding, J. T.; Louis, A. A. Hydrodynamic interactions and Brownian forces in colloidal suspensions: Coarse-graining over time and length scales. *Physical Review E* **2006**, *74* (3), 031402.
- (17) Benayad, Z.; von Bülow, S.; Stelzl, L. S.; Hummer, G. Simulation of FUS Protein Condensates with an Adapted Coarse-Grained Model. *Journal of Chemical Theory and Computation* **2021**, *17* (1), 525-537.
- (18) Guenza, M. Thermodynamic consistency and other challenges in coarse-graining models. *The European Physical Journal Special Topics* **2015**, *224* (12), 2177-2191.
- (19) Faller, R. Automatic coarse graining of polymers. *Polymer* **2004**, *45* (11), 3869-3876.

- (20) Milano, G.; Müller-Plathe, F. Mapping Atomistic Simulations to Mesoscopic Models: A Systematic Coarse-Graining Procedure for Vinyl Polymer Chains. *The Journal of Physical Chemistry B* **2005**, *109* (39), 18609-18619.
- (21) Nikoubashman, A.; Lee, V. E.; Sosa, C.; Prud'homme, R. K.; Priestley, R. D.; Panagiotopoulos, A. Z. Directed Assembly of Soft Colloids through Rapid Solvent Exchange. *ACS Nano* **2016**, *10* (1), 1425-1433.
- (22) Rubinstein, M.; Colby, R. Polymer Physics Oxford University Press. *New York* **2003**.
- (23) Guevara-Carrion, G.; Vrabc, J.; Hasse, H. Prediction of self-diffusion coefficient and shear viscosity of water and its binary mixtures with methanol and ethanol by molecular simulation. *The Journal of Chemical Physics* **2011**, *134* (7).
- (24) Kirkwood, J. G.; Riseman, J. The Intrinsic Viscosities and Diffusion Constants of Flexible Macromolecules in Solution. *The Journal of Chemical Physics* **2004**, *16* (6), 565-573.
- (25) Aust, C.; Kröger, M.; Hess, S. Structure and Dynamics of Dilute Polymer Solutions under Shear Flow via Nonequilibrium Molecular Dynamics. *Macromolecules* **1999**, *32* (17), 5660-5672.
- (26) Schuster, B. S.; Dignon, G. L.; Tang, W. S.; Kelley, F. M.; Ranganath, A. K.; Jahnke, C. N.; Simpkins, A. G.; Regy, R. M.; Hammer, D. A.; Good, M. C.; et al. Identifying sequence perturbations to an intrinsically disordered protein that determine its phase-separation behavior. *Proceedings of the National Academy of Sciences* **2020**, *117* (21), 11421-11431.
- (27) Nott, Timothy J.; Petsalaki, E.; Farber, P.; Jervis, D.; Fussner, E.; Plochowitz, A.; Craggs, T. D.; Bazett-Jones, David P.; Pawson, T.; Forman-Kay, Julie D.; et al. Phase Transition of a Disordered Nuage Protein Generates Environmentally Responsive Membraneless Organelles. *Molecular Cell* **2015**, *57* (5), 936-947.
- (28) Ghosh, A.; Kota, D.; Zhou, H.-X. Shear relaxation governs fusion dynamics of biomolecular condensates. *Nature Communications* **2021**, *12* (1), 5995.
- (29) Jawerth, L.; Fischer-Friedrich, E.; Saha, S.; Wang, J.; Franzmann, T.; Zhang, X.; Sachweh, J.; Ruer, M.; Ijavi, M.; Saha, S.; et al. Protein condensates as aging Maxwell fluids. *Science* **2020**, *370* (6522), 1317-1323.
- (30) Ibraheem, A.; Wade, M. B.; Samuel, R. C.; Mina, F.; Anurag, S.; Anne, B.; Rohit, V. P.; Tanja, M.; Priya, R. B. Sequence-encoded grammars determine material properties and physical aging of protein condensates. *bioRxiv* **2023**, 2023.04.06.535902.
- (31) Kota, D.; Zhou, H.-X. Macromolecular Regulation of the Material Properties of Biomolecular Condensates. *The Journal of Physical Chemistry Letters* **2022**, *13* (23), 5285-5290.
- (32) Shiv, R.; Cristobal Garcia, G.; Mayur, B.; Azamat, R.; Benjamin, S. S.; Kristi, L. K.; Jeetain, M. Expanding the molecular language of protein liquid-liquid phase separation. *bioRxiv* **2023**, 2023.03.02.530853.
- (33) Galvanetto, N.; Ivanović, M. T.; Chowdhury, A.; Sottini, A.; Nüesch, M. F.; Nettels, D.; Best, R. B.; Schuler, B. Extreme dynamics in a biomolecular condensate. *Nature* **2023**.
- (34) Semenov, A. N.; Rubinstein, M. Thermoreversible Gelation in Solutions of Associative Polymers. 1. Statics. *Macromolecules* **1998**, *31* (4), 1373-1385.
- (35) Tanaka, F.; Edwards, S. F. Viscoelastic properties of physically crosslinked networks. 1. Transient network theory. *Macromolecules* **1992**, *25* (5), 1516-1523.
- (36) Rubinstein, M.; Semenov, A. N. Thermoreversible Gelation in Solutions of Associating Polymers. 2. Linear Dynamics. *Macromolecules* **1998**, *31* (4), 1386-1397.

Reviewers' Comments:

Reviewer #1:

Remarks to the Author:

The authors have addressed all of my comments and concerns satisfactorily.

Reviewer #2:

Remarks to the Author:

The authors made some changes in response to the previous round of comments, but the changes do not address the root of my concerns, and therefore those concerns largely remain intact. Let me again highlight the issues raised.

1. Regarding the 10,000-fold gap in calculated and experimental viscosities, the authors' defense seems to be: a) "none of the CG models" can predict viscosities well; b) we have admitted this limitation; c) the reviewer's concern was caused by a confusion about the procedure used. Neither a) nor b) is much of a justification, and my concern was not due to a confusion about their procedure. The real issue is that CG models (and the implicit treatment of solvent) may fundamentally miss the actual physics behind viscosities of biomolecular condensates. Key determinants likely include: i) strengths and lifetimes of interaction networks; ii) ruggedness of the energy landscapes; iii) mediation by water and co-solvent molecules. CG models may well miss the relative weights of these factors. While an artificial scaling may bring the numerical value to the right order of magnitude, it may not correctly capture the intricate balance of the key determinants and thus miss the correct physical picture.

Indeed, my concern is borne out by a number of recent studies where all-atom explicit MD simulations were used to generate very interesting physical insights on the free energies and dynamics of biomolecular condensates (e.g., PMID: 33597515; PMID: 37468629). I'm not convinced that these physical insights can be captured well by CG models.

2. Regarding the failure of the viscopillary model, based on very limited data on surface tension, the authors very bravely concluded that "The viscopillary model does poorly when involving condensates made of components that all have folded regions (SH3 domain+lysozyme). Thus, the viscopillary model seems to be a reasonable choice for estimating the surface tension of condensates formed by fully disordered proteins but not necessarily for those involving folded domains." The truth is that both the amount data on surface tension and our understanding of viscoelasticity are far too limited to draw any meaningful conclusion at this time. As a case in point, in a recent study of condensates formed by fully disordered proteins (plus co-solvent), the measured surface tension differs from that inferred from the viscopillary model by 100-fold (PMID: 3764580).

In short, the physical insight from CG models is limited, especially in light of recent all-atom MD simulations. If taken out of context, one may even draw misleading conclusions.

Response to Reviews and Summary of Changes Made

Manuscript #: NCOMMS-23-21774A

Reviewers' Comments to Author:

Reviewer #1 (Remarks to the Author):

The authors have addressed all of my comments and concerns satisfactorily.

Response: We are glad to hear this and thank the reviewer again for providing constructive feedback which helped improve the manuscript.

Reviewer #2 (Remarks to the Author):

1a. Regarding the 10,000-fold gap in calculated and experimental viscosities, the authors' defense seems to be: a) "none of the CG models" can predict viscosities well; b) we have admitted this limitation; c) the reviewer's concern was caused by a confusion about the procedure used. Neither a) nor b) is much of a justification, and my concern was not due to a confusion about their procedure. The real issue is that CG models (and the implicit treatment of solvent) may fundamentally miss the actual physics behind viscosities of biomolecular condensates. Key determinants likely include: i) strengths and lifetimes of interaction networks; ii) ruggedness of the energy landscapes; iii) mediation by water and co-solvent molecules. CG models may well miss the relative weights of these factors. While an artificial scaling may bring the numerical value to the right order of magnitude, it may not correctly capture the intricate balance of the key determinants and thus miss the correct physical picture.

Indeed, my concern is borne out by a number of recent studies where all-atom explicit MD simulations were used to generate very interesting physical insights on the free energies and dynamics of biomolecular condensates (e.g., PMID: 33597515; PMID: 37468629). I am not convinced that these physical insights can be captured well by CG models.

Response: We are sorry to hear that the reviewer still feels that their concerns were not addressed in the revised manuscript. By reading carefully through the previous round of comments and the comments above, it appears that the reviewer's concerns largely stem from their belief that coarse-grained (CG) models *may* miss the actual physics or *may well* miss the relative weights of different factors. This unfounded criticism is further elaborated in terms of many (un)related issues

that pertain to the thermodynamics and dynamics (strengths of interaction networks, ruggedness of the energy landscape, mediation by water and co-solvent molecules) of self-assembly.

First, it is not clear at all what the reviewer’s specific scientific concern is about the measurements reported in our manuscript. The reviewer has failed to acknowledge the progress made using the CG model used in our work in terms of understanding the thermodynamics of phase separation in numerous previous studies,¹⁻⁹ which in our view renders the concerns about thermodynamic aspects mute. Of course, the community continues to improve the existing models, but we do not believe any of these issues will directly impact or invalidate the conclusions reported in our manuscript. As an example (out of many) of the predictive power of CG models, we reproduce the information from a recently published paper.¹⁰ Most importantly, no fitting parameters were used or system-specific tuning of CG model parameters was done to make quantitative comparisons between the experimental and simulation results in **Fig. R1**.

Fig. R1. Liquid-liquid phase separation (LLPS) of pHP1α in the presence of peptide ligands. Normalized supernatant pHP1α concentration of LLPS with and without peptide. Experimental data are shown as color symbols, while the simulated data are depicted as white symbols. The dashed lines represent the experimental ratio that is computed if LLPS does not occur. Each data point is normalized to the supernatant concentration of pHP1α in the absence of peptide.

Furthermore, it is worth highlighting that many of the previous insights in the literature have come from models that are significantly simpler (coarser) than the ones we used.¹¹ See for example the extensive works¹²⁻¹⁷ based on phenomenological on-lattice models within a sticker-spacer framework to make conclusions regarding several issues of interest in the condensate community. In fact, the results of our work have already been cited by a work from Prof. Schweizer’s group¹⁸ which develops a microscopic dynamic theory to explain the dynamics and rheology within condensates formed by sticker-spacer polymer models and how they relate to

internal condensate structure, highlighting that the conclusions of our work are general and are not dependent on the CG models used.

Second, the reviewer incorrectly labels our timescale mapping scheme as an artificial scaling. In fact, our results show that the mapping based on dilute phase chain diffusion can also work for describing the appropriate timescales inside the condensate, which should be taken as further evidence that our models *do not* miss any actual physics. CG models have been shown to work exceptionally well for describing and predicting the equilibrium/static features of IDPs and biomolecular condensates. Further, there is extensive literature from "classical" polymer physics which show that one can reproduce the correct dynamical behavior if one is mindful of the actual coarse-graining steps.

In the end, we strongly believe that the reviewer's bias against CG models is driving the criticism of our work. Moreover, scientific progress is not made only based on results without any *minor* underlying concerns about the current state-of-the-art methodology, but rather based on the appropriate use of these methods, while pushing the future frontiers by developing new techniques that address these concerns. In the spirit of a meaningful scientific exchange, the reviewer should have provided more serious and formal arguments against our paper's conclusions or methods, which they failed to articulate in our view.

Changes made to the manuscript: We have modified the last sentence of the discussion section (**Page 23**) to cite the recent theoretical work that references the results of this work as follows: "We believe that the molecular insights provided in this work will aid in further theoretical developments¹⁸ as well as computational and experimental investigations of diverse sequence features in dictating condensate dynamics, leading to a comprehensive molecular language for the material properties of biomolecular condensates."

1b. Indeed, my concern is borne out by a number of recent studies where all-atom explicit MD simulations were used to generate very interesting physical insights on the free energies and dynamics of biomolecular condensates (e.g., PMID: 33597515; PMID: 37468629). I am not convinced that these physical insights can be captured well by CG models.

Response: As the reviewer is likely familiar, the success of the all-atom approaches cited above relies on the ability to set up a condensed phase using the type of CG models we use in the current manuscript. Otherwise, it is currently impossible to create a relaxed dense phase configuration by just using a fully atomistic model. We discussed this in the very first study published by us on this topic.¹⁹

Most importantly, none of the papers cited by the reviewer provide quantitative (or even qualitative) information on the materials properties reported in our work such as viscosity,

interfacial tension, etc. due to the limited timescales and poor sampling achievable in these atomistic simulations, as we pointed out already in our previous response. We are left to wonder if the reviewer is asking us to do the impossible!

2. Regarding the failure of the viscocapillary model, based on very limited data on surface tension, the authors very bravely concluded that "The viscocapillary model does poorly when involving condensates made of components that all have folded regions (SH3 domain + lysozyme). Thus, the viscocapillary model seems to be a reasonable choice for estimating the surface tension of condensates formed by fully disordered proteins but not necessarily for those involving folded domains." The truth is that both the amount data on surface tension and our understanding of viscoelasticity are far too limited to draw any meaningful conclusion at this time. As a case in point, in a recent study of condensates formed by fully disordered proteins (plus co-solvent), the measured surface tension differs from that inferred from the viscocapillary model by 100-fold (PMID: 37645809).

Response: As pointed out by the reviewer, the amount of experimental data available is currently limited, thereby making it difficult to make general conclusions. Nonetheless, the excellent agreement between the simulation results from a physics-based CG model and the available experimental data (direct estimate) cannot be simply ignored as fortuitous.

As far as the applicability of the viscocapillary model is concerned, our conclusions were derived from a scientifically rigorous process based on the available data. We thank the reviewer for bringing the new preprint²⁰ (posted on August 23, 2023) to our attention, which looks at a more specialized system of ATP-mediated phase separation. To account for these new observations, we have now added the following caveat in the revised manuscript regarding our limited understanding of viscoelasticity and the role of environmental variables such as salt and ATP in modulating the behavior of condensates. Our study and the analysis of the applicability of the viscocapillary model should play an important role in the development of our fundamental understanding regarding the material properties of biomolecular condensates.

Changes made to the manuscript: We have added the following sentence in the Methods section (**Page 30**) of the manuscript: "However, given that the spatiotemporal evolution of condensates is only beginning to be explored,^{13, 21-25} we advise caution in inferring the surface tension of condensates through indirect measurements *via* the viscocapillary model, particularly when environmental variables such as salt and ATP²⁰ play a role in regulating the thermodynamics and dynamics of biomolecular condensates."

In short, the physical insight from CG models is limited, especially in light of recent all-atom MD simulations. If taken out of context, one may even draw misleading conclusions.

Response: We thank the reviewer for taking the time to provide critical feedback even though we disagree with their conclusions about the CG models. We believe the readers will be able to draw their conclusions based on this exchange, thereby minimizing any chances of causing confusion.

REFERENCES

- (1) Dignon, G. L.; Zheng, W.; Best, R. B.; Kim, Y. C.; Mittal, J. Relation between single-molecule properties and phase behavior of intrinsically disordered proteins. *Proceedings of the National Academy of Sciences* **2018**, *115* (40), 9929-9934.
- (2) Dignon, G. L.; Zheng, W.; Kim, Y. C.; Best, R. B.; Mittal, J. Sequence determinants of protein phase behavior from a coarse-grained model. *PLOS Computational Biology* **2018**, *14* (1), e1005941.
- (3) Schuster, B. S.; Dignon, G. L.; Tang, W. S.; Kelley, F. M.; Ranganath, A. K.; Jahnke, C. N.; Simpkins, A. G.; Regy, R. M.; Hammer, D. A.; Good, M. C.; et al. Identifying sequence perturbations to an intrinsically disordered protein that determine its phase-separation behavior. *Proceedings of the National Academy of Sciences* **2020**, *117* (21), 11421-11431.
- (4) Dignon, G. L.; Zheng, W.; Kim, Y. C.; Mittal, J. Temperature-Controlled Liquid–Liquid Phase Separation of Disordered Proteins. *ACS Central Science* **2019**, *5* (5), 821-830.
- (5) Krainer, G.; Welsh, T. J.; Joseph, J. A.; Espinosa, J. R.; Wittmann, S.; de Csilléry, E.; Sridhar, A.; Toprakcioglu, Z.; Gudiškytė, G.; Czekalska, M. A.; et al. Reentrant liquid condensate phase of proteins is stabilized by hydrophobic and non-ionic interactions. *Nature Communications* **2021**, *12* (1), 1085.
- (6) Espinosa, J. R.; Joseph, J. A.; Sanchez-Burgos, I.; Garaizar, A.; Frenkel, D.; Collepardo-Guevara, R. Liquid network connectivity regulates the stability and composition of biomolecular condensates with many components. *Proceedings of the National Academy of Sciences* **2020**, *117* (24), 13238-13247.
- (7) Tesei, G.; Schulze, T. K.; Crehuet, R.; Lindorff-Larsen, K. Accurate model of liquid–liquid phase behavior of intrinsically disordered proteins from optimization of single-chain properties. *Proceedings of the National Academy of Sciences* **2021**, *118* (44), e2111696118.
- (8) Das, S.; Muthukumar, M. Microstructural Organization in α -Synuclein Solutions. *Macromolecules* **2022**, *55* (11), 4228-4236.
- (9) Ranganathan, S.; Dasmeh, P.; Furniss, S.; Shakhnovich, E. Phosphorylation sites are evolutionary checkpoints against liquid–solid transition in protein condensates. *Proceedings of the National Academy of Sciences* **2023**, *120* (20), e2215828120.
- (10) Her, C.; Phan, T. M.; Jovic, N.; Kapoor, U.; Ackermann, B. E.; Rizuan, A.; Kim, Young C.; Mittal, J.; Debelouchina, Galia T. Molecular interactions underlying the phase separation of HP1 α : role of phosphorylation, ligand and nucleic acid binding. *Nucleic Acids Research* **2022**, *50* (22), 12702-12722.
- (11) Pappu, R. V.; Cohen, S. R.; Dar, F.; Farag, M.; Kar, M. Phase Transitions of Associative Biomacromolecules. *Chemical Reviews* **2023**, *123* (14), 8945-8987.
- (12) Bremer, A.; Farag, M.; Borchers, W. M.; Peran, I.; Martin, E. W.; Pappu, R. V.; Mittag, T. Deciphering how naturally occurring sequence features impact the phase behaviours of disordered prion-like domains. *Nature Chemistry* **2022**, *14* (2), 196-207.

- (13) Ibraheem, A.; Wade, M. B.; Samuel, R. C.; Mina, F.; Anurag, S.; Anne, B.; Rohit, V. P.; Tanja, M.; Priya, R. B. Sequence-encoded grammars determine material properties and physical aging of protein condensates. *bioRxiv* **2023**, 2023.2004.2006.535902.
- (14) Martin, E. W.; Holehouse, A. S.; Peran, I.; Farag, M.; Incicco, J. J.; Bremer, A.; Grace, C. R.; Soranno, A.; Pappu, R. V.; Mittag, T. Valence and patterning of aromatic residues determine the phase behavior of prion-like domains. *Science* **2020**, *367* (6478), 694-699.
- (15) Farag, M.; Borchers, W. M.; Bremer, A.; Mittag, T.; Pappu, R. V. Phase separation of protein mixtures is driven by the interplay of homotypic and heterotypic interactions. *Nature Communications* **2023**, *14* (1), 5527.
- (16) Seim, I.; Posey, A. E.; Snead, W. T.; Stormo, B. M.; Klotsa, D.; Pappu, R. V.; Gladfelter, A. S. Dilute phase oligomerization can oppose phase separation and modulate material properties of a ribonucleoprotein condensate. *Proceedings of the National Academy of Sciences* **2022**, *119* (13), e2120799119.
- (17) Wang, J.; Choi, J.-M.; Holehouse, A. S.; Lee, H. O.; Zhang, X.; Jahnel, M.; Maharana, S.; Lemaitre, R.; Pozniakovskiy, A.; Drechsel, D.; et al. A Molecular Grammar Governing the Driving Forces for Phase Separation of Prion-like RNA Binding Proteins. *Cell* **2018**, *174* (3), 688-699.e616.
- (18) Shi, G.; Schweizer, K. S. Theory of the center-of-mass diffusion and viscosity of microstructured and variable sequence copolymer liquids. *arXiv preprint arXiv:2310.04524* **2023**.
- (19) Zheng, W.; Dignon, G. L.; Jovic, N.; Xu, X.; Regy, R. M.; Fawzi, N. L.; Kim, Y. C.; Best, R. B.; Mittal, J. Molecular Details of Protein Condensates Probed by Microsecond Long Atomistic Simulations. *The Journal of Physical Chemistry B* **2020**, *124* (51), 11671-11679.
- (20) Divya, K.; Ramesh, P.; Huan-Xiang, Z. ATP Mediates Phase Separation of Disordered Basic Proteins by Bridging Intermolecular Interaction Networks. *bioRxiv* **2023**, DOI: 10.1101/2023.08.20.554035.
- (21) Jawerth, L.; Fischer-Friedrich, E.; Saha, S.; Wang, J.; Franzmann, T.; Zhang, X.; Sachweh, J.; Ruer, M.; Ijavi, M.; Saha, S.; et al. Protein condensates as aging Maxwell fluids. *Science* **2020**, *370* (6522), 1317-1323.
- (22) Fisher, R. S.; Elbaum-Garfinkle, S. Tunable multiphase dynamics of arginine and lysine liquid condensates. *Nature Communications* **2020**, *11* (1), 4628.
- (23) Ghosh, A.; Kota, D.; Zhou, H.-X. Shear relaxation governs fusion dynamics of biomolecular condensates. *Nature Communications* **2021**, *12* (1), 5995.
- (24) Galvanetto, N.; Ivanović, M. T.; Chowdhury, A.; Sottini, A.; Nüesch, M. F.; Nettels, D.; Best, R. B.; Schuler, B. Extreme dynamics in a biomolecular condensate. *Nature* **2023**.
- (25) Alshareedah, I.; Moosa, M. M.; Pham, M.; Potoyan, D. A.; Banerjee, P. R. Programmable viscoelasticity in protein-RNA condensates with disordered sticker-spacer polypeptides. *Nature Communications* **2021**, *12* (1), 6620.

Reviewers' Comments:

Reviewer #3:

Remarks to the Author:

In this work the authors, by means of computer simulations, test the hypothesis of whether condensate viscoelastic material properties can be inferred from single-protein calculations using multiple sequence variations of a model sequence of E-K, and two naturally occurring proteins: LAF-1-RGG, and the N-terminal domain of DDX4. They show that indeed, that is the case. To that end, they show correlations of single-protein radius of gyration against viscosity, diffusion, and surface tension. Moreover, they clearly show how charge segregation enhances condensate density, viscosity, and surface tension. The paper is well written, and in general clear. Apart from a few methodological concerns, that should be clarified and improved by the authors, the paper seems to be technically correct and a valuable contribution for wide community. However, one of my main concerns is whether the novelty is high, or some of the results were somehow expected. Nevertheless, regarding this last point, I must recall, that sometimes despite a result being "expected", it needs to be proved, and this paper indeed shows evidence of what it claims, hence it deserves credit. Below, I comment on several aspects that need to be addressed before is considered for publication in Nature Communications.

Major concerns:

1) What's novel in this work vs. previous works in the field using CG models for studying the effect of sequence patterning in protein LLPS? There are a few, which I am sure the authors know well, such as Statt et al., JCP, 2020 or Das et al., PCCP, 2018 investigating sequence patterning. It is true that in previous works the comparison of sequence patterning, or charge segregation (somehow analogous although still relatively different), goes against condensate stability whereas here it is related to condensate viscoelasticity, but it is kind of expected that more dense condensates with stronger network connectivity display larger viscosities as shown previously by Blazquez et al., Adv. Sci., 2023. An important effort highlighting what is new in this article with respect to previous articles should be emphasized in the main text.

2) The role of counterions and water for this kind of highly charged IDPs forming condensates can be highly relevant. This model does not account for how both water and counterions may alter the viscosity predictions for these condensates. Despite this kind of calculations cannot be performed through all-atom calculations (obviously), it would be really helpful if for at least a few sequences, it could be done through CG models including explicit solvent and ions (i.e., Martini or CG extensions for residue-resolution protein models with explicit water as in Benayad et al., JCTC, 2020). A comparison of how a very segregated sequence of E-K and a relatively homogeneously distributed sequence of E-K also show similar trends as obtained here when CG solvent and ions are explicitly considered would be significantly robust. In principle, condensates formed by sequences of 50 residues each should be feasible to simulate and sample with a model like that for verifying that the conclusions of the paper hold.

3) The way in which viscosity has been calculated is fairly complex, non-trivial, and it might be susceptible to significantly large uncertainties due to convergence reasons (almost as any viscosity calculation for polymer models). It would be very convincing if for a couple of

systems, the Green-Kubo method (as discussed by Tejedor et al., JPCB, 2023) would be applied for verifying that consistent results are obtained between different approaches, given that viscosity is one of the main magnitudes discussed across the article.

Minor concerns:

0) The way in which the simulation timescale is mapped into the experimental timescale is not fully clear. In the “match” between D_{exp} and D_{sim} , it is not clear what means when it gives values of the order of 10^{-9} seconds. How has this been calculated? The authors should provide a table indicating all the values for each step of this conversion since it is a key point of one of the main claims in the paper.

1) Sentences like these: *“Moreover, we found that the computed material properties from our simulations were quantitatively comparable with experimental measurements of charge-rich protein condensates”* along the discussion section should be avoided unless it is clearly stated that a rescaling factor has been applied. It might be misleading not to properly acknowledge that only qualitative predictions can be provided through CG models. Hence, although the proposed rescaling is a really interesting approach for comparing with experiments, it has to be clearly stated when discussed.

2) The way in which Figs. S7, S8 and S9 have been computed is not 100% clear. The authors should provide more details on how this has been computed.

3) Where does the 1.97 factor from the viscocapillarity expression come from? It is not evident. What does it mean? It might not be obvious for a broad audience, so please provide further details of its meaning and references of where to know more about this approximation. Importantly the disclaimer of this approach for inferring the surface tension from experimental measurements should be also stated in the main text.

4) Can be also provided further references and details on the relation: $PMF = kT \ln(RDF)$? It definitely looks meaningful, but its derivation or at least reference to it needs to be provided.

5) A priori, it should be equivalent to compute the diffusion through the MSD of just the central residue, or multiple residues across every protein replica (in principle it is more recommendable using central residues as the authors did). At very long timescales, it should not depend even on the location of the chosen residue across the sequence. Can the authors please show whether that is the case for their calculations? It can be a good consistency check. The MSD figures look quite good, but commenting on this aspect mentioned here might be useful for future computational studies following this interesting work.

0) When reading the main text, it is not easy to spot (if given) at which temperature all simulations were carried out. In the Methods is indeed provided. Could this be included in the main text? Moreover, at the beginning of the article, it is said that the most homogeneously charged variants of the model (E-K) sequence do not undergo LLPS. It would be very interesting for this article to include the stability dependence (i.e., phase diagrams, or critical temperature) of all the variants studied with respect to nSCD. It may require a large set of additional simulations, but it would truly add a very important angle to the article. At least for

some of them (i.e., 4-6 variants per protein: LAF-1, DDX4, and E-K one) it would be highly recommended.

Overall, the article is very interesting, but it requires further revision before being considered for publication in Nature Communications.

Response to Reviews and Summary of Changes Made

Manuscript #: NCOMMS-23-21774B-Z

Reviewers' Comments to Author:

Reviewer #1 (Remarks to the Author):

In this work the authors, by means of computer simulations, test the hypothesis of whether condensate viscoelastic material properties can be inferred from single-protein calculations using multiple sequence variations of a model sequence of E-K, and two naturally occurring proteins: LAF-1-RGG, and the N-terminal domain of DDX4. They show that indeed, that is the case. To that end, they show correlations of single-protein radius of gyration against viscosity, diffusion, and surface tension. Moreover, they clearly show how charge segregation enhances condensate density, viscosity, and surface tension. The paper is well written, and in general clear. Apart from a few methodological concerns, that should be clarified and improved by the authors, the paper seems to be technically correct and a valuable contribution for the wide community. However, one of my main concerns is whether the novelty is high, or some of the results were somehow expected. Nevertheless, regarding this last point, I must recall, that sometimes despite a result being “expected”, it needs to be proved, and this paper indeed shows evidence of what it claims, hence it deserves credit. Below, I comment on several aspects that need to be addressed before is considered for publication in Nature Communications.

Response: We thank the reviewer for their encouraging and constructively detailed assessment of our work. We second the reviewer’s comment that we have shown sufficient evidence for one of the most important findings of this work, which is the ability to infer the condensate material properties from the measurement of single-chain protein conformations. As the reviewer points out, our findings should be a valuable contribution to a broader community working on biomolecular condensates, phase separation, proteins, and polymeric systems. Our point-by-point responses to the comments can be found below.

Major concerns:

1) What’s novel in this work vs. previous works in the field using CG models for studying the effect of sequence patterning in protein LLPS? There are a few, which I am sure the authors know

well, such as Statt et al., JCP, 2020 or Das et al., PCCP, 2018 investigating sequence patterning. It is true that in previous works the comparison of sequence patterning, or charge segregation (somehow analogous although still relatively different), goes against condensate stability whereas here it is related to condensate viscoelasticity, but it is kind of expected that more dense condensates with stronger network connectivity display larger viscosities as shown previously by Blazquez et al., Adv. Sci., 2023. An important effort highlighting what is new in this article with respect to previous articles should be emphasized in the main text.

Response: We thank the reviewer for pointing out some of the previous articles^{1, 2} that have investigated the influence of sequence patterning on the thermodynamic properties of condensates formed by intrinsically disordered proteins (IDPs). The primary distinction between those articles and our work is that they only studied the influence of IDP sequence patterning on their ability to phase separate through the characterization of phase diagrams and critical temperatures, while we uncovered the sequence-dependent changes in the material properties of IDP condensates. Besides the articles that the reviewer has referenced,^{1, 2} there are several other studies on the sequence-dependent phase separation of IDPs,³⁻¹⁶ but the material state of protein condensates and its dependence on sequence is only beginning to be explored.¹⁷⁻²³ We agree with the reviewer that some of the results of this work may have been expected based on the known thermodynamic behavior, but they still need to be verified rigorously which we did in this work. Also, there are surprising findings in this work that are not expected or cannot be derived from the published literature. To this end, we investigated the following key aspects regarding the material properties of IDP condensates: whether different mesoscopic material properties can be predicted from one another with changes in the sequence charge patterning, whether the microscopic contact dynamics provide insights into the different condensate material properties, and whether the interactions that govern the single-chain IDP conformations also influence their condensate material properties.

First, we agree with the reviewer that more concentrated condensates should exhibit slower dynamics (*i.e.*, smaller diffusion coefficients or higher viscosities), as is shown in the recent study by Blazquez *et al.*²⁰ This behavior likely originates from the long-lived molecular contacts which engender slower dynamics, in line with the associative polymer theories²⁴⁻²⁶ which we have referenced in the Discussion section of the manuscript. However, one of our key findings is that a wide range of material properties of IDP condensates (diffusion coefficient, viscosity, and surface tension) showed nearly identical rate of change with increasing charge segregation for both the model and natural proteins, despite their very different sequence composition (**Figs. 2 and 3**). In addition, we found that these properties depend on each other (**insets in Figs. 2 and 3**), indicating the possibility of predicting these quantities from one another as well as the ability to modulate them simultaneously through sequence charge patterning without relying on external conditions such as temperature and salt concentration. Our results will serve as a baseline to further investigate the use of different charge patterning parameters to predict the material properties of protein condensates over a broader sequence space, similar to the article² referenced

by the reviewer, which discusses the usefulness and limitations of such parameters for predicting the phase separation propensity of IDPs.

Second, a recent study by Galvanetto *et al.*²⁷ in the *Nature* journal indicated that the local molecular contacts within a biomolecular condensate are highly transient (*i.e.*, multivalent interactions), which would facilitate efficient biochemical reactions, despite exhibiting a high viscosity. However, to the best of our knowledge, it has not been shown whether there exists a universal correlation between the *microscopic* contact dynamics and the *mesoscopic* material properties with changes in sequence charge patterning. We quantitatively showed that the contact lifetime between the oppositely charged residues and the different condensate material properties (in the dense-phase as well as at the interface) fall on a master curve, exhibiting a strong correlation, for disordered proteins with different sequence characteristics such as the ones investigated in this work (**Fig. 4**). This observation points to the possibility of using short equilibrium simulations for characterizing the contact dynamics to infer the IDP condensate material properties.

Third, a pivotal discovery of our work, for which sufficient evidence is provided as the reviewer acknowledges, is that the similar molecular interactions observed at the single-chain level and within the condensates of IDPs also lead to strong correlations between the single-chain size and their various condensate material properties (**Fig. 5**). Previously, such a correlation between the single-chain size and phase separation propensity was established by us and others,^{4, 28} which led to other studies investigating this aspect for diverse protein sequence features.⁹ It is now widely accepted that single-chain simulations of coil-to-globule transitions are powerful tools to estimate the phase behavior of protein sequences. We believe that the correspondence between the single-chain size and the condensate material properties shown in this work will prove to be an invaluable tool for interpreting condensate dynamics. Our findings will stimulate further studies to probe the sequence-dependent material properties of biomolecular condensates, focusing on non-ionic interactions and hydrophobic interactions.

In addition to the above findings derived from our purely computational study, we also demonstrated the ability of coarse-grained (CG) models to allow for a quantitative comparison of the time-dependent quantities such as viscosity with those measured experimentally (**Fig. 3b**), when one is mindful of the timescales used in the coarse-graining steps (see our response to the 4th comment of the reviewer). Further, thanks to the reviewer, we have now demonstrated that our findings are not an artifact of the CG models used, by simulating the protein sequences using a widely-used Martini model that includes explicit solvent and ions²⁹ (see our response to the 2nd comment of the reviewer). In fact, the results of our work have already been cited by a recent study from Prof. Schweizer's group,³⁰ in which a microscopic dynamic theory to explain the dynamics and rheology within condensates formed by sticker-spacer polymer models and how they relate to internal condensate structure has been developed. This again highlights that the conclusions of our work are general and are not dependent on the CG models used.

Changes made to the manuscript: We have made changes to the Discussion section (**Pages 21-25**) of the manuscript, highlighting the key findings from our work in light of the literature articles on disordered protein condensates, as has been discussed in this response.

2) The role of counterions and water for this kind of highly charged IDPs forming condensates can be highly relevant. This model does not account for how both water and counterions may alter the viscosity predictions for these condensates. Despite this kind of calculations cannot be performed through all-atom calculations (obviously), it would be really helpful if for at least a few sequences, it could be done through CG models including explicit solvent and ions (i.e., Martini or CG extensions for residue-resolution protein models with explicit water as in Benayad et al., JCTC, 2020). A comparison of how a very segregated sequence of E-K and a relatively homogeneously distributed sequence of E-K also show similar trends as obtained here when CG solvent and ions are explicitly considered would be significantly robust. In principle, condensates formed by sequences of 50 residues each should be feasible to simulate and sample with a model like that for verifying that the conclusions of the paper hold.

Response: We agree with the reviewer that the role of counterions and water could play an important role in dictating the dynamics and rheology of charge-rich IDP condensates. To address this aspect, we simulated all 15 E-K sequences using the latest Martini force field version 3, that models the solvent and ion particles explicitly.²⁹ To characterize the diffusion coefficient of the protein chains and the viscosity in the dense phase, we first simulated the E-K sequences in a slab geometry (**Fig. R1a**) for a total duration of 18 μ s. This duration allowed for the protein and water to reach their equilibrium dense phase concentrations ρ , after which we cut out the dense phase section to simulate it in a cuboid simulation box (**Fig. R1a**). The simulations in a cuboid box were done at a salt concentration of 100 mM to emulate the conditions of our previous CG simulations with the hydrophathy scale (HPS) model.^{3,31} The details of the Martini simulations and the characterization of the dynamical and rheological properties from these simulations are provided in the Methods section of the manuscript. Similar to the HPS model, we found that the perfectly alternating E-K variant with nSCD = 0 did not form a stable dense phase in the Martini model. For the other E-K sequences, we found that the dense phase concentration ρ for the protein increased with increasing nSCD (i.e., increasing charge segregation), similar to that seen in the HPS model (**Fig. R1b**). The absence of explicit solvent in the HPS model resulted in higher dense phase concentrations as compared to the Martini model. While the concentration of water decreased monotonically with increasing nSCD in the Martini model, it remained higher than the protein concentration for all E-K sequences, similar to that observed from the all-atom simulations of condensates formed by natural disordered proteins such as FUS low complexity domain and LAF1 RGG domain.³²

From the dense phase Martini simulation trajectories of the E-K sequences, we computed

Fig. R1. (a) Dense phase snapshot of Martini set-up for one of the E–K sequences in the presence of water (cyan) and ions (yellow) in a cuboid simulation box, which was cut from a slab geometry, for characterizing its dense phase diffusive and rheological properties. (b) Comparison of the dense phase concentration ρ for protein and water as a function of nSCD in the Martini model with the protein’s concentration in the HPS model for the E–K sequences. (c) Inversely proportional correlation between normalized diffusion coefficient D^* and normalized zero-shear viscosity η_0^* from the Martini and HPS models for the E–K sequences, with the solid line corresponding to the Stokes-Einstein type relation $D^* = 1/\eta_0^*$. (d) Correlation between η_0^* and normalized single-chain R_g^* from the Martini and HPS models for the E–K sequences. The solid line corresponds to the linear fit $\ln(\eta_0^*) = -4.98\ln(R_g^*) + 0.19$ with correlation coefficient $R^2 = 0.90$. The values of D^* , η_0^* , and R_g^* are obtained after normalizing D , η_0 , and R_g by those computed for the reference E–K sequence with nSCD = 0.017 in the HPS and Martini models. The symbol color, ranging from blue to red in (c) and (d), indicates increasing nSCD.

Fig. R2. (a) Mean square displacement $\text{MSD}(t)$ of the residues of a chain for the E–K sequences simulated using the Martini model. The slope value 1 indicates the diffusive regime. (b) Diffusion coefficient D is extracted from the plateau region observed at long times ($t \geq 5 \mu\text{s}$) where the relation $\text{MSD} = 6Dt$ holds. The line color, ranging from blue to red, indicates increasing nSCD.

Fig. R3. (a) Zero-shear viscosity η_0 is obtained from the plateau region of the Green-Kubo integral observed at long times ($t \geq 0.5 \mu\text{s}$) for the E–K sequences simulated using the Martini model. The line color, ranging from blue to red, indicates increasing nSCD.

the diffusion coefficient D from the mean square displacement (MSD) of the residues in a protein chain (**Fig. R2**) and the zero-shear viscosity η_0 (**Fig. R3**) from the Green-Kubo relation that is given in our response to the reviewer’s 3rd comment. We found that dynamics in the Martini model was ~ 2 orders of magnitude slower than that observed in the HPS model, when the respective intrinsic molecular dynamics (MD) timescale is used. For example, the values of η_0 for the most-well mixed sequence that phase separated (nSCD = 0.017) and the diblock sequence (nSCD = 1) from the Martini model were 46.23 mPa · s and 291.74 mPa · s, respectively, while those from the HPS model were 0.087 mPa · s and 1.315 mPa · s, respectively. Thus, the relevant timescales in both CG models need to be rescaled appropriately. When plotted as a normalized quantity, we found that D^* and η_0^* from both the Martini and HPS models closely followed the Stokes-Einstein relation $D^* = 1/\eta_0^*$ (**Fig. R1c**), highlighting that the changes in these quantities can be predicted from one another when the sequence charge patterning gets altered. Further, we found that η_0^* of the dense phase was strongly correlated with the single-chain radius of gyration R_g^* in both the Martini and HPS models (**Fig. R1d**). Our consistent observations from the Martini and HPS models provide evidence that our findings hold true irrespective of the models used.

Changes made to the manuscript: We have included **Fig. R1** as a new supplementary figure **Fig. S3** and the discussion of the Martini model throughout the manuscript (**Pages 1, 4, 8, 12, 21, 23, and 24**). We have also discussed about the MSD of the residues within a protein chain exhibiting the diffusive regime from which D was extracted (**Fig. R2** as **Fig. S24**) as well as the convergence of η_0 obtained from the Green-Kubo relation (**Fig. R3** as **Fig. S25**) in the Methods section (**Pages 34-36**) of the manuscript. Further, we have added the simulation details of the Martini model in the Methods section (**Pages 34-36**) of the manuscript.

3) The way in which viscosity has been calculated is fairly complex, non-trivial, and it might be susceptible to significantly large uncertainties due to convergence reasons (almost as any viscosity calculation for polymer models). It would be very convincing if for a couple of systems, the Green-Kubo method (as discussed by Tejedor et al., JPCB, 2023) would be applied for verifying that consistent results are obtained between different approaches, given that viscosity is one of the main magnitudes discussed across the article.

Response: We thank the reviewer for this concern regarding the convergence of the zero-shear viscosity η_0 data obtained from the nonequilibrium molecular dynamics (NEMD) simulations. In the previous version of the manuscript, we showed that the viscosity obtained from the NEMD simulations converged to the reported mean η_0 value at times shorter than the total simulation time for the investigated protein sequences (**Fig. S21**). To further justify the convergence of the reported η_0 , we computed it from equilibrium MD simulations using the Green-Kubo relation.^{33, 34}

$$\eta_0 = \int_0^\infty G(t) dt,$$

Fig. R4. (a) Shear stress relaxation modulus $G(t)$ for select E–K sequences. The solid lines at long time scales represent the fit to the Maxwell modes. (b) Comparison between zero shear viscosity η_0 for the E–K sequences obtained from the NEMD simulations and the Green-Kubo relation. In approach I, η_0 was computed based on the Green-Kubo relation by simply integrating $G(t)$. In approach II, η_0 was computed as per the procedure outlined by Tejedor *et al.*¹⁹

where $G(t)$ is the shear stress relaxation modulus. We measured $G(t)$ based on the autocorrelation of the pressure tensor components P_{ab} (**Fig. R4a**),^{21,35}

$$G(t) = \frac{V}{5k_B T} [\langle P_{xy}(0)P_{xy}(t) \rangle + \langle P_{xz}(0)P_{xz}(t) \rangle + \langle P_{yz}(0)P_{yz}(t) \rangle] \\ + \frac{V}{30k_B T} [\langle N_{xy}(0)N_{xy}(t) \rangle + \langle N_{xz}(0)N_{xz}(t) \rangle + \langle N_{yz}(0)N_{yz}(t) \rangle],$$

where V is the volume of the simulation box, k_B is the Boltzmann constant, and $N_{ab} = P_{aa} - P_{bb}$ is the normal stress difference. We computed η_0 in two different ways: the first approach involved obtaining the measurements by simply integrating $G(t)$ of different E–K sequences, while for the other set of measurements, we followed the approach outlined in the recent article by Tejedor *et al.*,¹⁹ which the reviewer has pointed out. In the second approach, to remove the influence of typical noisy behavior of $G(t)$ in the terminal regime at long time scales, we first fitted the values of $G(t)$ beyond the time after which the intramolecular oscillations have decayed to a series of Maxwell modes ($G_i \exp(-t/\tau_i)$ with $i = 1 \dots 4$) equidistant in logarithmic time (**Fig. R4a**).³⁶ We then obtained η_0 as the sum of numerical integration at short times and analytical integration based on the fitted data at long times (**Fig. R4b**). We found that the η_0 values obtained based on the Green-Kubo relation are in excellent quantitative agreement with those obtained from the NEMD simulations, providing evidence that the viscosity values reported in this work are an accurate

representation of the terminal flow characteristics of the IDP chains in the dense phase.

Changes made to the manuscript: We have included the figure discussed in this response as a new supplementary figure (**Fig. R4** as **Fig. S22**) and the associated discussion highlighting the consistency of the reported viscosity results from the NEMD simulations and the Green-Kubo method in the Methods section (**Pages 28 and 29**) of the manuscript.

Minor concerns:

4) The way in which the simulation timescale is mapped into the experimental timescale is not fully clear. In the “match” between D_{exp} and D_{sim} , it is not clear what means when it gives values of the order of 10^{-9} seconds. How has this been calculated? The authors should provide a table indicating all the values for each step of this conversion since it is a key point of one of the main claims in the paper.

Response: We have described our approach for mapping the simulation and experimental timescales at the single-chain level in the Methods section of the manuscript. To make it clearer, we have created a table (**Table R1**), describing the calculation of the new timescales in detail. Our description in the Methods section along with the table should help the readers of the manuscript in following the steps taken in arriving at the new timescales.

Protein sequence	D_{MD} based on intrinsic MD timescale (m^2/s)	D_{sim} in simulation units ($\sqrt{\epsilon\sigma^2/m}$ or σ^2/τ)	D_{exp} (m^2/s)	New timescale τ (s) from $D_{\text{sim}} = D_{\text{exp}}$
E–K	3.897×10^{-7}	27.99	1.571×10^{-10}	1.782×10^{-9}
LAF1	1.385×10^{-7}	9.07	0.539×10^{-10}	1.683×10^{-9}
DDX4	0.982×10^{-7}	6.45	0.867×10^{-10}	0.744×10^{-9}

Table R1. We first computed the diffusion coefficient D_{MD} of a single-chain based on the intrinsic MD timescale ($\tau = 48.89$ fs) for the E–K, LAF1, and DDX4 sequences (second column). Note that $T = 300$ K was used for the E–K and DDX4 sequences, while $T = 280$ K was used for the LAF1 sequences. Next, we represented the diffusion coefficient in simulation units D_{sim} (third column) by dividing D_{MD} by $\sqrt{\epsilon\sigma^2/m}$ (or σ^2/τ) for the E–K, LAF1, and DDX4 sequences, respectively. Average mass m of each sequence and length scale $\sigma = 10^{-10}$ m were used in computing the values of $\sqrt{\epsilon\sigma^2/m}$. We also calculated the experimentally expected diffusion coefficient D_{exp} of a single-chain in water as per the Stokes-Einstein relation (fourth column). Finally, we matched D_{sim} with D_{exp} to obtain new timescales τ (fifth column) that would yield the same single-chain diffusion coefficient in both simulations and experiments for the E–K, LAF1, and DDX4 sequences.

Changes made to the manuscript: We added a new supplementary table (**Table R1** as **Table S3**), describing how we arrived at the new timescales for rescaling the diffusivities and viscosities obtained from our simulations, which should be complementary to the discussion in the Methods section (**Pages 31-33**) of the manuscript.

5) Sentences like these: “*Moreover, we found that the computed material properties from our simulations were quantitatively comparable with experimental measurements of charge-rich protein condensates*” along the discussion section should be avoided unless it is clearly stated that a rescaling factor has been applied. It might be misleading not to properly acknowledge that only qualitative predictions can be provided through CG models. Hence, although the proposed rescaling is a really interesting approach for comparing with experiments, it has to be clearly stated when discussed.

Response: We have modified the highlighted sentence, implementing the reviewer’s suggestion, in the revised manuscript.

Changes made to the manuscript: We have rephrased the sentence in the Discussion section (**Page 24**) of the manuscript as follows: “We found that the material properties of the condensates computed from our simulations were quantitatively comparable with experimental measurements of charge-rich protein condensates, when the timescales were rescaled based on the single-chain dynamics (a standard practice in soft matter physics; see Methods).”

6) The way in which Figs. S7, S8, and S9 have been computed is not 100% clear. The authors should provide more details on how this has been computed.

Response: We first computed the probability of a pair of residues i and j in a sequence of length N to be in contact as $P = \langle n_{ij} \rangle$, with $n_{ij} = 1$ if the distance between the pair was less than the cut-off radius of 1.5σ , and $n_{ij} = 0$ if not. This procedure resulted in a $N \times N$ matrix of contact probabilities. Instead of summing up all the contact probabilities $\sum P$ in each column that would result in an one-dimensional vector of length N , we resorted to computing $\sum P$ for each residue in the sequence based on the following five classifications: (1) charged residue i with all oppositely charged residues j in the sequence, (2) charged residue i with all other like charged residues j in the sequence, (3) charged residue i with all uncharged residues j in the sequence, (4) uncharged residue i with all other uncharged residues j in the sequence, and (5) uncharged residue i with all charged residues j in the sequence. As an example, for the LAF1 sequence ($N = 168$) in which there are 44 charged residues and 124 uncharged residues, the above classification resulted in a $\sum P$ vector of length $(3 \times 44) + (2 \times 124) = 380$ for the single-chain and in the dense phase, which we have plotted in **Fig. S10**.

Changes made to the manuscript: We have added detailed description on how $\sum P$ was

computed for the single-chain and in the dense phase (**Figs. S9-S11**) in the Methods section (**Pages 30 and 31**) of the manuscript.

7) Where does the 1.97 factor from the viscopillarity expression come from? It is not evident. What does it mean? It might not be obvious for a broad audience, so please provide further details of its meaning and references of where to know more about this approximation. Importantly the disclaimer of this approach for inferring the surface tension from experimental measurements should be also stated in the main text.

Response: The factor of 1.97 in the viscopillarity model comes from the observation that the time evolution of the edge-to-edge distance of two fusing viscous droplets, initially with equal radius R (Stokes model), quantitatively followed the stretched exponential functional form, with stretching exponent $\beta = 1.5$ and fusion time $\tau_f = 1.97\tau_{vc} = 1.97 \frac{\eta R}{\gamma}$, where τ_{vc} , η and γ corresponds to the viscopillarity time, viscosity, and surface tension of the droplets.^{23, 37, 38} In other words, the stretched exponential parameters $\beta = 1.5$ and $\tau_f \left(\frac{\gamma}{\eta R}\right) = 1.97$ was merely observed to fit the numerical solution of the Stokes model,³⁸ but had no theoretical basis.

Changes made to the manuscript: We have added a sentence in the Methods section (**Pages 33 and 34**) of the manuscript, referring the readers to the appropriate references^{23, 37, 38} regarding the details of the viscopillarity model. Also, we have already stated in the Methods section (**Page 34**) of the manuscript, advising caution in using the viscopillarity model to indirectly infer surface tension of condensates.

8) Can be also provided further references and details on the relation: $PMF = -kT \ln(RDF)$? It definitely looks meaningful, but its derivation or at least reference to it needs to be provided.

Response: We thank the reviewer for noting the missing reference, and have now referred the readers to the statistical mechanics book by David Chandler,³⁴ in which the derivation of the relation $PMF = -k_B T \ln(RDF)$ can be found.

Changes made to the manuscript: We have added the reference³⁴ in the Results section (**Page 15**) of the manuscript, where the $PMF = -k_B T \ln(RDF)$ is first introduced.

9) A priori, it should be equivalent to compute the diffusion through the MSD of just the central residue, or multiple residues across every protein replica (in principle it is more recommendable using central residues as the authors did). At very long timescales, it should not depend even on the location of the chosen residue across the sequence. Can the authors please show whether that is the case for their calculations? It can be a good consistency check. The MSD figures look quite good, but commenting on this aspect mentioned here might be useful for future computational

studies following this interesting work.

Response: We thank the reviewer for raising this concern regarding the computation of diffusion coefficient D from the mean square displacement (MSD) of the residues in a chain. In the previous version of the manuscript, we showed that the values of D , computed from the regime where $\text{MSD} \propto t$, are identical irrespective of whether they are measured from the MSD of the chain’s inner residues or from the MSD of the chain’s center of mass for the E–K, LAF1, and DDX4 sequences (**Figs. R5a, S16a, and S17a**). This is because, for sufficiently long times $t \gg \tau_R$, τ_R being the Rouse relaxation time of the entire chain, the MSD of the residues becomes identical to the MSD of the chain, *i.e.*, MSD of the chain’s inner residues relative to its center of mass g_2 plateaus at long times (**Figs. R5b, S16b, and S17b**).³⁹ Based on reviewer’s suggestion, we computed the MSD of only the end monomers of a chain and found that they overlap with the MSD of the chain’s inner residues at long times for the E–K, LAF1, and DDX4 sequences (**Figs. R5a, S16a, and S17a**), further highlighting that our simulations have converged.

Fig. R5. (a) Mean square displacement $\text{MSD}(t)$ of the inner residues of a chain (solid lines), the center of mass of a chain (dashed lines), and the end monomers of a chain (dotted lines) in the dense phase for select E–K sequences ($n\text{SCD} = 0.017, 0.200, \text{ and } 1.000$). The slope values 2, 0.5, and 1 indicate ballistic, sub-diffusive, and diffusive regimes, respectively. Inner residues correspond to those in the middle of a chain after excluding 20 residues on either end of the sequence. (b) MSD of the inner residues of a chain relative to the center of mass $g_2(t)$ in the dense phase for select E–K sequences. The line color, ranging from blue to red, indicates increasing $n\text{SCD}$.

Changes made to the manuscript: We have included new supplementary figures (**Figs. S15-S17**) showing that the MSD of the inner monomers, the end monomers, and the center of mass of a chain are the same at long times for the E–K, LAF1, and DDX4 sequences. We have

also added sentences discussing that the location of the residues in a sequence does not affect the measurement of the translational diffusion coefficient D of the protein chains in the Methods section (Page 27) of the manuscript.

10) When reading the main text, it is not easy to spot (if given) at which temperature all simulations were carried out. In the Methods is indeed provided. Could this be included in the main text? Moreover, at the beginning of the article, it is said that the most homogeneously charged variants of the model (E-K) sequence do not undergo LLPS. It would be very interesting for this article to include the stability dependence (i.e., phase diagrams, or critical temperature) of all the variants studied with respect to nSCD. It may require a large set of additional simulations, but it would truly add a very important angle to the article. At least for some of them (i.e., 4-6 variants per protein: LAF-1, DDX4, and E-K one) it would be highly recommended.

Response: As per the reviewer’s suggestion, we have stated the temperature at which the E–K, LAF1, and DDX4 sequences were simulated before discussing the results of this work. Also, we investigated the influence of charge patterning on the stability of condensates by characterizing the phase diagrams for select LAF1, and DDX4 sequences (Fig. R6). We found that the condensates were stable over a wide range of temperatures, as indicated by higher critical temperature, with increasing charge segregation. We chose not to characterize the phase diagram and critical temperature of the E–K sequences, as these are computationally expensive simulations and are already available in the literature,^{4,28} known to show similar trends as the LAF1 and DDX4 charge variants (Fig. R6).

Fig. R6. Phase diagram for select (a) LAF1 (nSCD = 0.003, 0.010, 0.131, and 0.394) and (b) DDX4 (nSCD = 0.002, 0.021, 0.194, and 0.478) sequence variants. The open symbols correspond to the estimated critical temperature and critical density of the sequence variants. The line color, ranging from blue to red, indicates increasing nSCD. Lines are guide to the eye only.

Changes made to the manuscript: We have now added sentences describing the temperatures at which the material properties of the condensates formed by the E–K, LAF1, and DDX4 sequences were obtained in the Results section (**Page 6**) of the manuscript. To justify our choice of the temperatures, we also included a new supplementary figure (**Fig. R6** as **Fig. S1**) and discussion regarding the stability of condensates formed by the charge-rich protein sequences studied in this work in the Results section (**Page 7**) of the manuscript. Further, we have added details regarding the simulations carried out to obtain the phase diagrams and the fits to the simulation data for estimating the critical temperature and critical density in the Methods section (**Page 30**) of the manuscript.

Additional changes made to the manuscript:

1. Since the reviewer comments required a substantial amount of additional simulations and analysis, we have included two new authors to the manuscript: Beata Szała-Mendyk and Shiv Rekhi as third and fourth authors, respectively.
2. We have added 5 new supplementary figures and a new table, because of which the supplemental figure numbers and table numbers have changed from the previous version of the manuscript.

REFERENCES

- (1) Statt, A.; Casademunt, H.; Brangwynne, C. P.; Panagiotopoulos, A. Z. Model for disordered proteins with strongly sequence-dependent liquid phase behavior. *The Journal of Chemical Physics* **2020**, *152* (7).
- (2) Das, S.; Amin, A. N.; Lin, Y.-H.; Chan, H. S. Coarse-grained residue-based models of disordered protein condensates: utility and limitations of simple charge pattern parameters. *Physical Chemistry Chemical Physics* **2018**, *20* (45), 28558-28574.
- (3) Dignon, G. L.; Zheng, W.; Kim, Y. C.; Best, R. B.; Mittal, J. Sequence determinants of protein phase behavior from a coarse-grained model. *PLOS Computational Biology* **2018**, *14* (1), e1005941.
- (4) Dignon, G. L.; Zheng, W.; Best, R. B.; Kim, Y. C.; Mittal, J. Relation between single-molecule properties and phase behavior of intrinsically disordered proteins. *Proceedings of the National Academy of Sciences* **2018**, *115* (40), 9929-9934.
- (5) Schuster, B. S.; Dignon, G. L.; Tang, W. S.; Kelley, F. M.; Ranganath, A. K.; Jahnke, C. N.; Simpkins, A. G.; Regy, R. M.; Hammer, D. A.; Good, M. C.; et al. Identifying sequence perturbations to an intrinsically disordered protein that determine its phase-separation behavior. *Proceedings of the National Academy of Sciences* **2020**, *117* (21), 11421-11431.
- (6) Shiv, R.; Cristobal Garcia, G.; Mayur, B.; Azamat, R.; Benjamin, S. S.; Kristi, L. K.; Jeetain, M. Expanding the molecular language of protein liquid-liquid phase separation. *bioRxiv* **2023**, 2023.2003.2002.530853.

- (7) Krainer, G.; Welsh, T. J.; Joseph, J. A.; Espinosa, J. R.; Wittmann, S.; de Csilléry, E.; Sridhar, A.; Toprakcioglu, Z.; Gudiškytė, G.; Czekalska, M. A.; et al. Reentrant liquid condensate phase of proteins is stabilized by hydrophobic and non-ionic interactions. *Nature Communications* **2021**, *12* (1), 1085.
- (8) Tesei, G.; Schulze, T. K.; Crehuet, R.; Lindorff-Larsen, K. Accurate model of liquid–liquid phase behavior of intrinsically disordered proteins from optimization of single-chain properties. *Proceedings of the National Academy of Sciences* **2021**, *118* (44), e2111696118.
- (9) Bremer, A.; Farag, M.; Borchers, W. M.; Peran, I.; Martin, E. W.; Pappu, R. V.; Mittag, T. Deciphering how naturally occurring sequence features impact the phase behaviours of disordered prion-like domains. *Nature Chemistry* **2022**, *14* (2), 196-207.
- (10) Farag, M.; Borchers, W. M.; Bremer, A.; Mittag, T.; Pappu, R. V. Phase separation of protein mixtures is driven by the interplay of homotypic and heterotypic interactions. *Nature Communications* **2023**, *14* (1), 5527.
- (11) Martin, E. W.; Holehouse, A. S.; Peran, I.; Farag, M.; Incicco, J. J.; Bremer, A.; Grace, C. R.; Soranno, A.; Pappu, R. V.; Mittag, T. Valence and patterning of aromatic residues determine the phase behavior of prion-like domains. *Science* **2020**, *367* (6478), 694-699.
- (12) Her, C.; Phan, T. M.; Jovic, N.; Kapoor, U.; Ackermann, B. E.; Rizuan, A.; Kim, Young C.; Mittal, J.; Debelouchina, Galia T. Molecular interactions underlying the phase separation of HP1 α : role of phosphorylation, ligand and nucleic acid binding. *Nucleic Acids Research* **2022**, *50* (22), 12702-12722.
- (13) Nott, Timothy J.; Petsalaki, E.; Farber, P.; Jervis, D.; Fussner, E.; Plochowietz, A.; Craggs, T. D.; Bazett-Jones, David P.; Pawson, T.; Forman-Kay, Julie D.; et al. Phase Transition of a Disordered Nuage Protein Generates Environmentally Responsive Membraneless Organelles. *Molecular Cell* **2015**, *57* (5), 936-947.
- (14) Seim, I.; Posey, A. E.; Snead, W. T.; Stormo, B. M.; Klotsa, D.; Pappu, R. V.; Gladfelter, A. S. Dilute phase oligomerization can oppose phase separation and modulate material properties of a ribonucleoprotein condensate. *Proceedings of the National Academy of Sciences* **2022**, *119* (13), e2120799119.
- (15) Wang, J.; Choi, J.-M.; Holehouse, A. S.; Lee, H. O.; Zhang, X.; Jahnel, M.; Maharana, S.; Lemaitre, R.; Pozniakovsky, A.; Drechsel, D.; et al. A Molecular Grammar Governing the Driving Forces for Phase Separation of Prion-like RNA Binding Proteins. *Cell* **2018**, *174* (3), 688-699.e616.
- (16) Joseph, J. A.; Reinhardt, A.; Aguirre, A.; Chew, P. Y.; Russell, K. O.; Espinosa, J. R.; Garaizar, A.; Collepardo-Guevara, R. Physics-driven coarse-grained model for biomolecular phase separation with near-quantitative accuracy. *Nature Computational Science* **2021**, *1* (11), 732-743.
- (17) Jawerth, L.; Fischer-Friedrich, E.; Saha, S.; Wang, J.; Franzmann, T.; Zhang, X.; Sachweh, J.; Ruer, M.; Ijavi, M.; Saha, S.; et al. Protein condensates as aging Maxwell fluids. *Science* **2020**, *370* (6522), 1317-1323.

- (18) Ibraheem, A.; Wade, M. B.; Samuel, R. C.; Mina, F.; Anurag, S.; Anne, B.; Rohit, V. P.; Tanja, M.; Priya, R. B. Sequence-encoded grammars determine material properties and physical aging of protein condensates. *bioRxiv* **2023**, 2023.2004.2006.535902.
- (19) Tejedor, A. R.; Collepardo-Guevara, R.; Ramírez, J.; Espinosa, J. R. Time-Dependent Material Properties of Aging Biomolecular Condensates from Different Viscoelasticity Measurements in Molecular Dynamics Simulations. *The Journal of Physical Chemistry B* **2023**, *127* (20), 4441-4459.
- (20) Blazquez, S.; Sanchez-Burgos, I.; Ramirez, J.; Higginbotham, T.; Conde, M. M.; Collepardo-Guevara, R.; Tejedor, A. R.; Espinosa, J. R. Location and Concentration of Aromatic-Rich Segments Dictates the Percolating Inter-Molecular Network and Viscoelastic Properties of Ageing Condensates. *Advanced Science* **2023**, *10* (25), 2207742.
- (21) Tejedor, A. R.; Sanchez-Burgos, I.; Estevez-Espinosa, M.; Garaizar, A.; Collepardo-Guevara, R.; Ramirez, J.; Espinosa, J. R. Protein structural transitions critically transform the network connectivity and viscoelasticity of RNA-binding protein condensates but RNA can prevent it. *Nature Communications* **2022**, *13* (1), 5717.
- (22) Fisher, R. S.; Elbaum-Garfinkle, S. Tunable multiphase dynamics of arginine and lysine liquid condensates. *Nature Communications* **2020**, *11* (1), 4628.
- (23) Ghosh, A.; Kota, D.; Zhou, H.-X. Shear relaxation governs fusion dynamics of biomolecular condensates. *Nature Communications* **2021**, *12* (1), 5995.
- (24) Semenov, A. N.; Rubinstein, M. Thermoreversible Gelation in Solutions of Associative Polymers. 1. Statics. *Macromolecules* **1998**, *31* (4), 1373-1385.
- (25) Rubinstein, M.; Semenov, A. N. Thermoreversible Gelation in Solutions of Associating Polymers. 2. Linear Dynamics. *Macromolecules* **1998**, *31* (4), 1386-1397.
- (26) Tanaka, F.; Edwards, S. F. Viscoelastic properties of physically crosslinked networks. 1. Transient network theory. *Macromolecules* **1992**, *25* (5), 1516-1523.
- (27) Galvanetto, N.; Ivanović, M. T.; Chowdhury, A.; Sottini, A.; Nüesch, M. F.; Nettels, D.; Best, R. B.; Schuler, B. Extreme dynamics in a biomolecular condensate. *Nature* **2023**.
- (28) Lin, Y.-H.; Chan, H. S. Phase Separation and Single-Chain Compactness of Charged Disordered Proteins Are Strongly Correlated. *Biophysical Journal* **2017**, *112* (10), 2043-2046.
- (29) Souza, P. C. T.; Alessandri, R.; Barnoud, J.; Thallmair, S.; Faustino, I.; Grünewald, F.; Patmanidis, I.; Abdizadeh, H.; Bruininks, B. M. H.; Wassenaar, T. A.; et al. Martini 3: a general purpose force field for coarse-grained molecular dynamics. *Nature Methods* **2021**, *18* (4), 382-388.
- (30) Shi, G.; Schweizer, K. S. Theory of the center-of-mass diffusion and viscosity of microstructured and variable sequence copolymer liquids. *arXiv preprint arXiv:2310.04524* **2023**.
- (31) Regy, R. M.; Thompson, J.; Kim, Y. C.; Mittal, J. Improved coarse-grained model for studying sequence dependent phase separation of disordered proteins. *Protein Science* **2021**, *30* (7), 1371-1379.

- (32) Zheng, W.; Dignon, G. L.; Jovic, N.; Xu, X.; Regy, R. M.; Fawzi, N. L.; Kim, Y. C.; Best, R. B.; Mittal, J. Molecular Details of Protein Condensates Probed by Microsecond Long Atomistic Simulations. *The Journal of Physical Chemistry B* **2020**, *124* (51), 11671-11679.
- (33) Zwanzig, R. Time-Correlation Functions and Transport Coefficients in Statistical Mechanics. *Annual Review of Physical Chemistry* **1965**, *16* (1), 67-102.
- (34) Chandler, D. *Introduction to Modern Statistical Mechanics*. Oxford University Press, Oxford, UK, 1987.
- (35) Ramírez, J.; Sukumaran, S. K.; Vorselaars, B.; Likhtman, A. E. Efficient on the fly calculation of time correlation functions in computer simulations. *The Journal of Chemical Physics* **2010**, *133* (15), 154103.
- (36) Rubinstein, M.; Colby, R. *Polymer Physics*. Oxford University Press, Oxford, UK, 2003.
- (37) Ghosh, A.; Zhou, H.-X. Determinants for Fusion Speed of Biomolecular Droplets. *Angewandte Chemie International Edition* **2020**, *59* (47), 20837-20840.
- (38) Martínez-Herrera, J. I.; Derby, J. J. Viscous Sintering of Spherical Particles via Finite Element Analysis. *Journal of the American Ceramic Society* **1995**, *78* (3), 645-649.
- (39) Nikoubashman, A.; Howard, M. P. Equilibrium Dynamics and Shear Rheology of Semiflexible Polymers in Solution. *Macromolecules* **2017**, *50* (20), 8279-8289.

Reviewers' Comments:

Reviewer #3:

Remarks to the Author:

The authors have successfully address my comments and concerns from the previous round of revision, and thus, I recommend the article for publication.